# Prefrontal coding of learned and inferred knowledge during REM and NREM sleep

Kareem Abdou [1,2,3,4], Masanori Nomoto [1,2,4,5], Mohamed H. Aly [1,2,4,6], Ahmed Z. Ibrahim [1,2,3,4], Kiriko Choko[1,2,4], Reiko Okubo-Suzuki[1,2,4], Shin-ichi Muramatsu [7,8] & Kaoru Inokuchi [1,2,4] ✉

Idling brain activity has been proposed to facilitate inference, insight, and innovative problem-solving. However, it remains unclear how and when the idling brain can create novel ideas. Here, we show that cortical offline activity is both necessary and sufficient for building unlearned inferential knowledge from previously acquired information. In a transitive inference paradigm, male C57BL/6J mice gained the inference 1 day after, but not shortly after, complete training. Inhibiting the neuronal computations in the anterior cingulate cortex (ACC) during post-learning either non-rapid eye movement (NREM) or rapid eye movement (REM) sleep, but not wakefulness, disrupted the inference without affecting the learned knowledge. In vivo $Ca^{2+}$ imaging suggests that NREM sleep organizes the scattered learned knowledge in a complete hierarchy, while REM sleep computes the inferential information from the organized hierarchy. Furthermore, after insufficient learning, artificial activation of medial entorhinal cortex-ACC dialog during only REM sleep created inferential knowledge. Collectively, our study provides a mechanistic insight on NREM and REM coordination in weaving inferential knowledge, thus highlighting the power of idling brain in cognitive flexibility.

Cognitive flexibility is a distinctive feature that is required for higher learning[1,2]. Inferential reasoning is a prominent property of cognitive flexibility since it relies on the flexible and systematic organization of existing knowledge[1,3]. Activation patterns of neurons in the medial prefrontal cortex (mPFC) have been suggested to represent inferential reasoning[4,5].

Experience-related neural representations are re-expressed during post-learning awake rest and sleep periods[6,7]. This neural replay has been proposed to consolidate memories[6,8–12]. Previous work has found that awake replay events that occur during a spatial task not only represent recent experiences but also novel spatial paths that might be confronted, which indicates that a whole spatial map has been

formed[13–15]. Targeted reactivation of an implicitly learned sequence memory during sleep enables the development of explicit awareness of that sequence[16]. Previous reports have shown an offline improvement in a word-pair association task following sleep[17–20].

Enhanced cognitive performance has been linked with offline activity during non-rapid eye movement (NREM)[21] and rapid eye movement (REM)[22] sleep, but the most pertinent sleep stage and the circuits involved are still unclear. REM and NREM sleep have unique neurophysiological processes that impact cognition[23,24]. REM sleep is characterized by ponto-geniculo-occipital waves, theta rhythm, high acetylcholine levels, and upregulation of plasticity-related genes[25]. Long-term potentiation is easily induced during REM sleep[26]. NREM

[1]Research Centre for Idling Brain Science, University of Toyama, Toyama 930-0194, Japan. [2]Department of Biochemistry, Graduate School of Medicine and Pharmaceutical Sciences, University of Toyama, Toyama, Japan. [3]Department of Biochemistry, Faculty of Pharmacy, Cairo University, Cairo 11562, Egypt. [4]CREST, Japan Science and Technology Agency (JST), University of Toyama, Toyama, Japan. [5]Japan Agency for Medical Research and Development (AMED), Tokyo, Japan. [6]Pharmacology Department, Faculty of Pharmacy, The British University in Egypt, Cairo 11837, Egypt. [7]Division of Neurological Gene Therapy, Centre for Open Innovation, Jichi Medical University, Tochigi 3290498, Japan. [8]Centre for Gene and Cell Therapy, The Institute of Medical Science, The University of Tokyo, Tokyo 1088639, Japan. ✉e-mail: inokuchi@med.u-toyama.ac.jp

sleep is characterized by regular occurrence of slow wave delta activity, high protein synthesis levels, and a rise in depotentiation-related genes and a drop in long-term potentiation-related genes[23,27–32]. Therefore, sleep stages differentially affect neuronal functions and may therefore modify cognitive capabilities. Furthermore, previous reports have proposed that NREM and REM sleep have differential contributions in several brain functions; for example, memory consolidation[33], visual learning[34], skills, and schemas[35]. Consequently, we hypothesize that NREM and REM sleep have distinct roles in the processing of learned and inferential information.

The aforementioned studies[13–16,21,22] highlight the involvement of online and offline brain activity in cognitive flexibility. However, the temporal, cellular, and circuit mechanisms underlying the emergence

of inference from chronologically separated, yet overlapping, experiences are still unclear.

## Results

### Randomizing existing knowledge is necessary for inference emergence

We developed a transitive inference paradigm in mice that assesses the ability to infer new information that was not learned before, based on previous knowledge of overlapping memories. First, food-restricted mice were allowed to explore and habituate to an arena consisting of five different contexts (Context A, B, C, D, and E) (Fig. 1a). These contexts were different in terms of geometry and/or floor texture and/or wall color and pattern. During the habituation period, mice did not

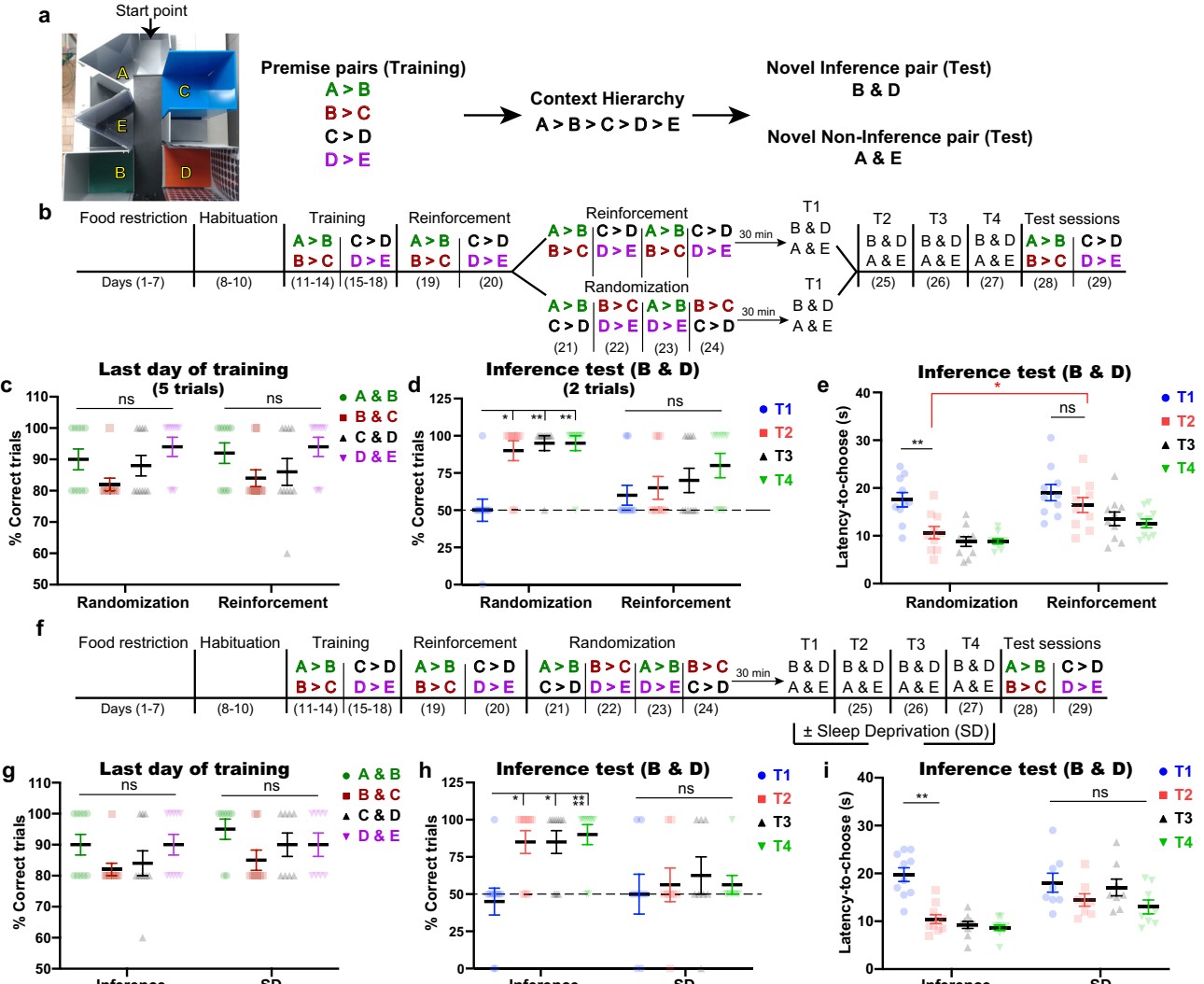

**Fig. 1 | The emergence of inference requires sleep after training.**
**a** Representative photo of the arena and the five different contexts with an arrow representing the starting point (left). The four premise pairs with the contexts' hierarchy (right); premise pairs were consistently color-coded across all figures to facilitate tracking each pair without confusion. **b** The behavioral schedule used to establish the transitive inference paradigm in mice. The contexts' hierarchy was as follows: A > B > C > D > E. The order of presentation of the premise pairs during the randomization stage is different across animals within the same group (See "Methods" section; Table 1). **c** Performance during the last day of training (day 14 for A > B and B > C; day 18 for C > D and D > E) for each premise pair was calculated as the percent of correct trials out of the total number of trials (5 trials).
**d**, **e** Performance during test sessions; the percent of correct trials in the inference test (2 trials) (**d**), the latency time to choose in the inference test (**e**). **f** Behavioral

protocol for the transitive inference paradigm; SD, sleep deprivation. **g** Percent of correct trials during the last day of training (day 14 for A > B and B > C; day 18 for C > D and D > E) for each premise pair. **h**, **i** Performance during test sessions; percent of correct trials during the inference test (**h**), latency time to choose during the inference test (**i**). T1, test 1; T2, test 2; T3, test 3; T4, test 4. n = 10 mice/group; n = 8 mice for the sleep deprivation (SD) group. Statistical comparisons were made using a two-way repeated-measures ANOVA with Tukey's multiple comparison test (**c**–**e**, **g**, **h**) and Holm–Sidak's test between groups (**e**). *$P < 0.05$; **$P < 0.01$; ****$P < 0.0001$; ns, not significant ($P > 0.05$). Data are presented as the mean ± standard error of the mean (s.e.m.). Experiments were independently repeated four times. Source data are provided as a Source Data file. Detailed statistics are shown in Supplementary Data 1.

show a significant innate preference to any context (Supplementary Fig. 1). Then, they were trained on a series of two-context discrimination trials called premise pairs (A > B, B > C, C > D, and D > E; where ">" means that the former context is more preferred than the latter one). Mice should prefer the context in which they were rewarded with a sucrose tablet. Every day, mice were trained on two premise pairs in two sessions, with each session consisting of five trials. During each trial, mice were put into the arena with only two opened contexts, while the other three contexts were closed (Fig. 1b). The correct choice requires 2 steps: (1) entering to the assigned correct context and (2) staying in the assigned context for a consecutive 10 s, then mice received a sucrose tablet (Supplementary Movie 1). After reaching the criterion for correct performance (80% correct trials) in the four premise pairs (Fig. 1c), mice were divided into two groups. The first group was subjected to reinforcement sessions only, while the second group received randomized training sessions in which each pair was presented with a different pair every day (Fig. 1b and Methods). Then, mice were subjected to inference (B & D) and non-inference (A & E) tests on the last day of training (T1) and in the subsequent 3 days (T2, T3, and T4) (Supplementary Movie 2). Contexts B and D had an equal valence during training as they were the preferred context in one premise pair (B > C and D > E, respectively), and were the avoided context in the other premise pair (A > B and C > D, respectively), such that the performance in the inference test would be purely due to inference from the hierarchy. On the other hand, context A was usually the preferred context, while context E was usually un-preferred; therefore, the performance during tests with the A and E contexts would not be due to the inference. Mice that received randomized training inferred correctly in all tests except for in T1, as shown by an increased number of correct trials and a decreased latency time to choose the correct context (Fig. 1d, e). On the other hand, mice that received reinforcement training only without randomization did not exhibit inference in any tests (Fig. 1d, e); however, they had successfully learned the original premise pairs (Fig. 1c), which indicates that knowledge of the premise pairs did not assure the emergence of inference. This showed the necessity of randomization during the acquisition of the premise pairs knowledge for gaining the inference. In both groups, the percentage of correct trials was significantly higher than the chance level (50%) in the non-inference test, and their memories of the original premises were intact (Supplementary Fig. 2a).

To examine the temporal contribution of the randomization stage in inference emergence, mice were trained on the same behavior protocol, but with only 2 days randomization (incomplete randomization) (Supplementary Fig. 3a). Mice successfully learned the premise pairs (Supplementary Fig. 3b); however, they did not make correct inference during testing sessions (Supplementary Fig. 3b). This result suggests that ensuring complete interaction between all premise pairs is necessary for inference emergence. Complete interaction between the learned pairs occurred after 4 days of randomized training (on day 24).

### The emergence of inference requires sleep after training
The inability to infer correctly in T1 and the expression of inference in T2 (Fig. 1d) suggest that inference does not arise online during the repeated training, but instead may require an incubation period or sleep following training. To examine which factor is critical for the inference process, we kept the incubation period intact (the same as before), but mice were sleep-deprived for 4 h directly after test sessions (Fig. 1f). Mice met the criterion for correct training performance, with a non-significant difference in the percent of correct trials across all premise pairs (Fig. 1g). Sleep deprivation blocked the inference; the correct performance of mice remained at chance level and their decision-making was delayed (Fig. 1h, i). However, sleep deprivation did not affect performance on the non-inference test or the original premises' tests (Supplementary Fig. 2b). This suggests that sleep after

the last randomized training is crucial for constructing the whole hierarchy of preferences and the subsequent development of inference, but that it is not essential for the A and E pair or for the maintenance of the original premises' representations. The inferential behavior that appeared in T2 in the inference group was not due to learning during T1 since mice were not provided with sucrose in T1, even when the correct choice was made (see "Methods" section).

To test whether the inference observed in T2 (Fig. 1d, h) was facilitated by T1 exposure in which the B and D pair was presented to mice, mice were exposed to the same behavior protocol without exposure to T1 (Supplementary Fig. 4a). After successful training phase (Supplementary Fig. 4b), mice inferred correctly during subsequent test sessions (Supplementary Fig. 4c, d). This result indicates that T1 exposure is not necessary for the emergence of inference during T2, which supports that inference is a higher-order process than a result of direct learning.

### Correct performance is observed with an incomplete hierarchy setting
Next, we tested the correct inference in an incomplete hierarchy setting. We trained a group of mice in the same arena, with the same number of trials, but with premise pairs that did not form a complete hierarchy (A > B, B > C, E > D, D > C; Supplementary Fig. 5a, b). Mice learned the premise pairs well and reached the criterion of correct performance (Supplementary Fig. 5c). In test sessions, mice did not prefer context B over context D due to lacking the complete hierarchy (Supplementary Fig. 5d, e), although mice memorized the original premise pairs well (Supplementary Fig. 5g). Since context A and context E were always preferred in the premise pairs training, mice did not prefer either of these during the non-inference test (Supplementary Fig. 5f). Altogether, the behavioral readout of correct performance was context B preference in the complete hierarchy setting, while it was a non-contextual preference in the incomplete hierarchy setting. These findings suggest that the behavioral manifestation of correct inference is contingent upon the presence or absence of a complete hierarchy.

### Prelimbic cortex activity is not required for inference emergence
The mPFC has been proposed to process the mental schema that is utilized to build an inference and may also compute the inferred outcomes[4,5,36-38]. However, these proposals were derived from lesion and functional brain imaging studies, and lack the causality for temporal and topographic specificity that are necessary for inference development. To directly unravel these ambiguities, we manipulated the activity of the mPFC sub-region, the prelimbic cortex (PL) during sleep and awake states. The PL of mice was injected with adeno-associated virus 9 (AAV9) encoding a light-sensitive neuronal silencer (ArchT 3.0) with enhanced yellow fluorescent protein under the control of calcium/calmodulin-dependent protein kinase II (CaMKII) to label excitatory neurons with ArchT 3.0 (Supplementary Fig. 6a). Then, mice were subjected to the transitive inference task, where optogenetic inhibition was performed during either wakefulness or sleep on the last training day and testing days (Supplementary Fig. 6a, b). Mice achieved the standard for correct performance in all premise pairs (Supplementary Fig. 6c, d). Inhibiting PL activity during sleep or wakefulness after the training did not affect the inference, non-inference test, or premise pairs memories (Supplementary Fig. 6e–h), which indicates that PL activities during sleep and awake states are not required for the formation of inference.

### Offline ACC computations are crucial for the emergence of inference
Next, we manipulated another mPFC sub-region, the anterior cingulate cortex (ACC). Mice received injections into the ACC of AAV-CaMKII-ArchT 3.0-eYFP to label excitatory neurons with ArchT 3.0 (Fig. 2a).

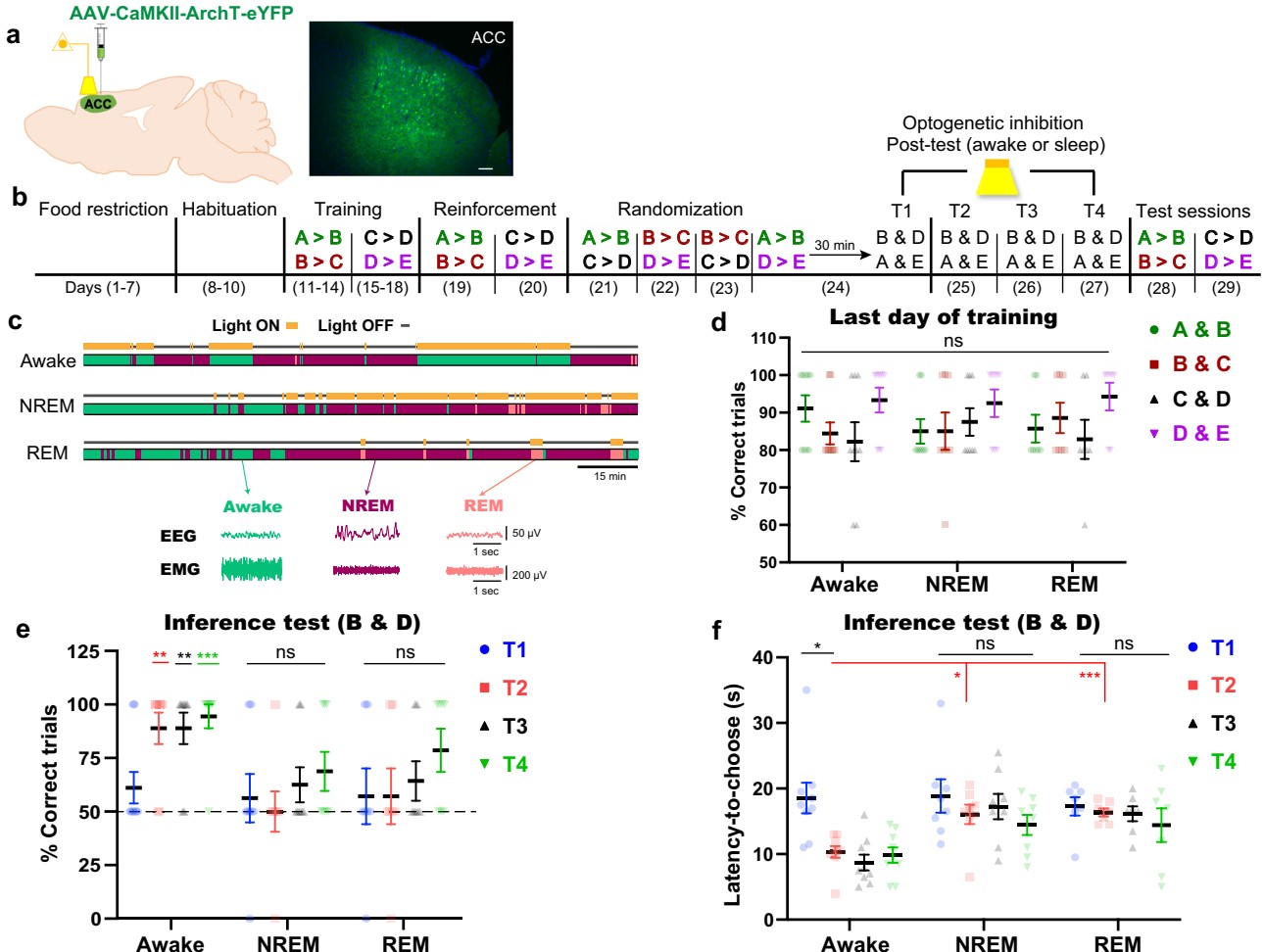

**Fig. 2 | ACC computations during sleep are crucial for the emergence of inference. a** Labeling excitatory neurons of the ACC with ArchT (left), and the expression of ArchT-eYFP (green) in the ACC (right). Blue, 4′,6-diamidino-2-phenylindole (DAPI) staining. Scale bar, 100 μm. **b** Behavioral schedule used to manipulate ACC activity during sleep and awake periods after test sessions. The order of presentation of the premise pairs during the randomization stage is different across animals within the same group (See "Methods" section; Table 1). **c** Diagram showing the state-specific manipulation (top) and representative electroencephalogram (EEG) and electromyography (EMG) (bottom) traces. Scale bar, 15 min. **d** Performance during the last day of training (day 14 for A > B and B > C; day 18 for C > D and D > E) for each premise pair. **e, f** Performance during test sessions;

percent of correct trials during the inference test (**e**), latency time to choose during the inference test (**f**). T1, test 1; T2, test 2; T3, test 3; T4, test 4. $n = 9$ mice for the awake group; $n = 8$ mice for the non-rapid eye movement (NREM) sleep group; $n = 7$ mice for the rapid eye movement (REM) sleep group. Statistical comparisons were made using two-way repeated-measures analysis of variance (ANOVA) with Tukey's multiple comparison test (**d**–**f**). In (**e**), the statistical significance denotes the comparison between performance relative to the chance level (50%). *$P < 0.05$; **$P < 0.01$; ***$P < 0.001$; ns, not significant ($P > 0.05$). Data are presented as the mean ± standard error of the mean (s.e.m.). The Experiment was independently repeated six times. Source data are provided as a Source Data file. Detailed statistics are shown in Supplementary Data 1.

Afterwards, mice completed the transitive inference task, where optogenetic inhibition was performed during either wakefulness, NREM sleep, or REM sleep on the last training day and testing days (Fig. 2b, c). Mice learned the premise pairs well, which was reflected by the high number of correct trials that reached the criterion for correct performance (Fig. 2d). Optogenetic inhibition of the ACC (Supplementary Fig. 7a) during wakefulness did not impact inference ability, as mice achieved a high correct performance during inference, non-inference, and premise pairs tests (Fig. 2e, f and Supplementary Fig. 2c). Conversely, mice that received optogenetic inhibition of the ACC either during NREM or REM sleep failed to infer correctly, and the percent of correct trials was not significantly different from chance level (Fig. 2e). Moreover, the latency to choose was significantly longer in the NREM and REM groups than that of the awake group (Fig. 2f). These data indicate that ACC computations are necessary for the evolution of inference ability during sleep, but not during wakefulness. However, this perturbation to ACC dynamics had no influence on the

non-inference pairs or the maintenance of premises knowledge (Supplementary Fig. 2c).

### Inference representations develop gradually, peaking in REM sleep

To examine how the offline brain organizes the previously acquired knowledge and builds novel inferential information, we tracked the ACC neuronal activity via Ca²⁺ imaging. Wild-type mice were injected with AAV9-Syn-janelia-GCaMP7 into the ACC (Fig. 3a) and a custom-built electroencephalogram/electromyography (EEG/EMG) 5-pin system was installed into the skull, as previously described[39], to record the neuronal dynamics during awake and sleep states (Fig. 3b). The same ACC neurons were tracked across sessions[6] along the transitive inference task (Fig. 3c–e), using an automated sorting system[40] to extract each neuron's calcium (Ca²⁺) activity which were normalized to z-scores (see "Methods" section for details). Offline stages either awake or sleep were differentiated according to the automatically

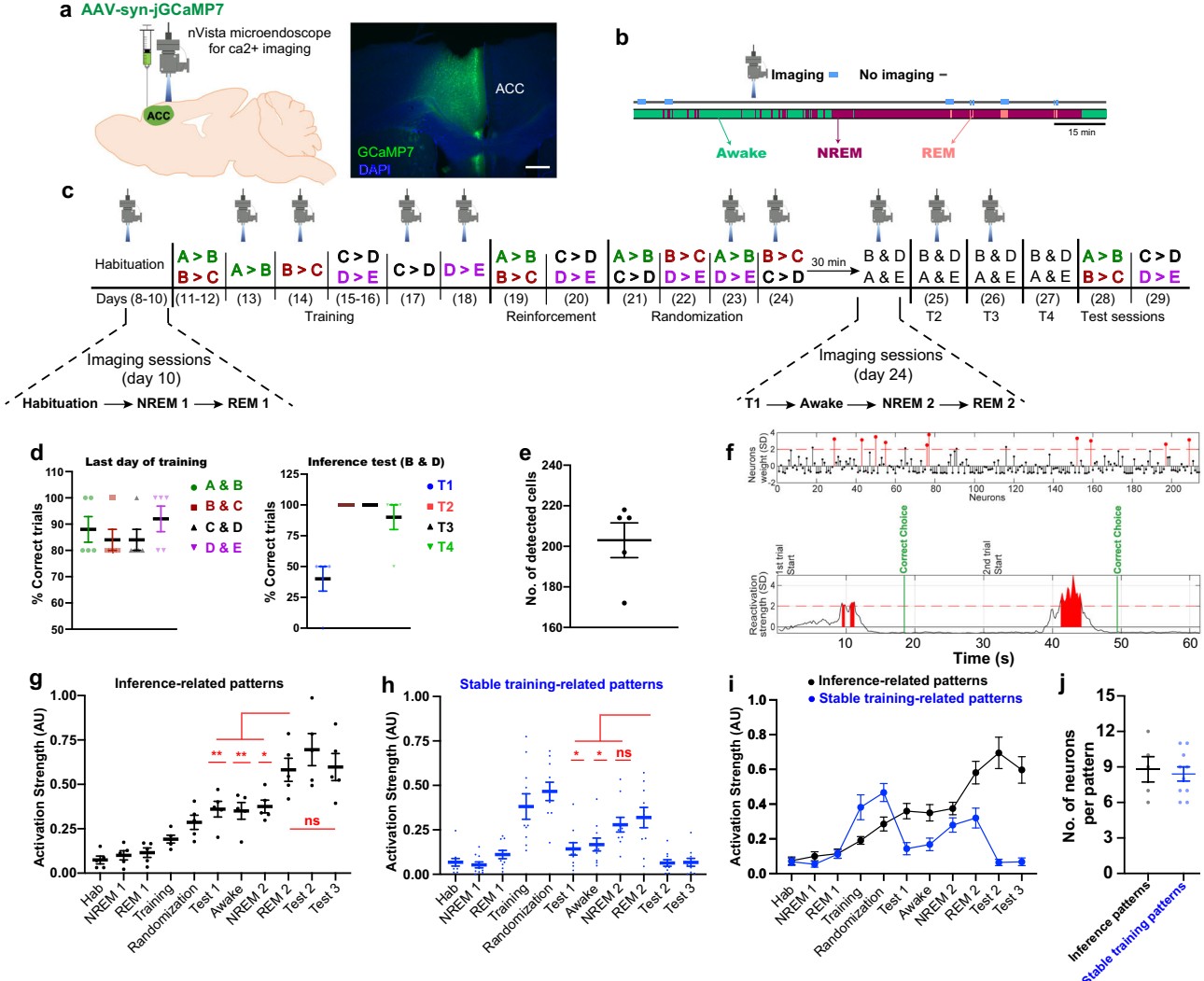

**Fig. 3 | Identifying inference-related ensembles that emerge gradually and become evident during REM sleep. a** Labeling of the ACC with GCaMP7 (left), and the expression of GCaMP7 (green) in the ACC (right). Blue, 4′,6-diamidino-2-phe-nylindole (DAPI) staining. Scale bar, 100 μm. **b** Diagram showing the state-specific imaging. **c** The behavioral schedule used to capture the Ca²⁺ transients across the task. On day 10, Ca²⁺ transients were recorded from 3 imaging sessions; habituation, NREM 1, and REM 1. On day 24, Ca²⁺ transients were recorded from 4 imaging sessions; T1, awake in home cage, NREM 2, and REM 2. During the training stage, Ca²⁺ transients were collected for each premise pair on a separate day (days 13, 14, 17, 18) to extract specific representations for a particular premise pair without interfering with the other pair. The order of presentation of the premise pairs during the randomization stage is different across animals within the same group (See "Methods" section; Table 1). **d** Performance during the last day of training (day 13 for A > B; day 14 for B > C; day 17 for C > D; day 18 for D > E) for each premise pair (left). Performance during test sessions (right). T1, test 1; T2, test 2; T3, test 3; T4, test 4. **e** Total number of detected cells in each mouse, *n* = 5 mice (**d**, **e**). **f** Example of ACC coactivity pattern (inference-related pattern) detected in T2 session.

The pattern is represented as a vector containing the contribution (weight) of each neuron's spiking to the coactivity defining that pattern. Neurons with a weight above 2 s.d. of the mean were referred to as members (Red) (top). The temporal appearance of the pattern with the behavior signature (just before the correct choice) (bottom). **g–i** Activation strength (z-scored) of inference-related patterns (**g, i**), of stable training-related patterns (**h, i**). **j** Number of neurons constituting both inference patterns and stable training patterns, *n* = 5 inference-related patterns (**g, i, j**); *n* = 10 Stable training-related patterns (**h, i, j**). Hab, habituation session; NREM 1, NREM sleep after last habituation session; REM 1, REM sleep after last habituation session; NREM 2, NREM sleep after T1; REM 2, REM sleep after T1. Statistical comparisons were made using one-way repeated-measures analysis of variance (ANOVA) with Dunnett's multiple comparison test (**g, h**). *\*P* < 0.05; *\*\*P* < 0.01; ns not significant (*P* > 0.05). Data are presented as the mean ± standard error of the mean (s.e.m.). The Experiment was independently repeated four times. Source data are provided as a Source Data file. Detailed statistics are shown in Supplementary Data 1.

enumerated EMG root mean square value (ERMS), EEG delta power (1–4 Hz) RMS (dRMS), and EEG theta power (6–9 Hz) RMS, as described previously[39,41]. Sleep stage discrimination was concluded based on the delta/theta ratio value (Fig. 3b; see "Methods" section for details).

Next, we employed a combination of principal component and independent component analyses (PCA/ICA) to identify the neuronal assembly patterns and track their activity over time (Fig. 3f, j). We isolated different synchronized patterns during the T2 session (Inference session) and tracked them over time. After that, we identified the

inference-related patterns according to 3 criteria. First, they appeared in both T2 and T3 sessions just before the correct choice (entering and waiting in context B). Second, they appeared in T2 with an activation strength significantly higher than their strength during the T1 session where the inferential behavior was absent. Third, they did not appear during (B & C) and (C & D) sessions to confirm that these patterns are not representing context B preference or context D avoidance.

Upon investigating the temporal emergence of inference-related patterns, we found that they started to emerge from the

randomization session and their activation strength increased gradually, peaking at REM sleep after the T1 session (Fig. 3g, i). The strength of inference-related patterns was comparable between T2, T3, and REM 2 sessions, which was significantly higher than their strength in the earlier sessions. This result indicates that significant inference representations emerge during REM sleep after randomized training. Although inference-related patterns appeared during the T1 session, their strength were not high enough to trigger inferential behavior in T1. They were evident enough to elicit inferential behavior in T2 and T3 sessions, suggesting that the level of pattern strength predicts the task performance.

On the other hand, we extracted neuronal patterns representing the premise pairs during training sessions (see "Methods" section). These patterns were divided into 3 groups; first, patterns activated significantly during initial training than during randomized training, named initial training-related patterns (Supplementary Fig. 8a). Second, patterns activated significantly during randomized training than during initial training, named randomized training-related patterns (Supplementary Fig. 8b). Third, patterns activated significantly during both initial and randomized training than during the earlier sessions, named stable training-related patterns (Stable "Tr, Rand") (Fig. 3h). The stable "Tr, Rand" patterns representing both initial and randomized training were evident during REM sleep more than the initial training-related patterns, suggesting a higher contribution of stable "Tr, Rand" patterns in inference emergence (Supplementary Fig. 8c). Unlike inference-related patterns which reactivated during REM 2 more than NREM 2, stable "Tr, Rand" patterns reactivated similarly during NREM 2 and REM 2 sessions (Fig. 3h, i). Furthermore, comparing the activation strength of inference-related patterns and stable "Tr, Rand" patterns across sessions suggests that representations of learned and inferred information might be orthogonal (Fig. 3i).

### Coactivity code underlies the emergence of inference representation

Understanding ACC computations during NREM 2 & REM 2 sleep sessions would explain the mechanism underlying the transitional increase of activation strength of inference-related patterns between NREM 2 and REM 2 (Fig. 3g), which predicts the inferential behavior. To address this aim, we assessed the coactivation index between different patterns by calculating the number of pairwise synchronized $Ca^{2+}$ activities within 200 ms between neurons constituting each pattern (Fig. 4a, b). Neurons contributing to the stable "Tr, Rand" patterns were coactivated significantly higher during NREM 2 than during REM 2 and awake sessions, representing a significant interaction (Fig. 4c). Furthermore, neurons belonging to inference-related patterns interacted with those belonging to stable "Tr, Rand" patterns significantly higher during REM 2 than during NREM 2 and awake sessions (Fig. 4c).

To examine the contribution of each premise pair (A&B, B&C, C&D, D&E) to the coactivity code, we extracted the patterns uniquely representing each premise pair and tracked the coactivity of their neurons. We counted the quadruple coactivation index between neurons representing the 4 individual premise pairs and found that they coactivated together during NREM 2 significantly more than during REM 2 session, which suggests organizing the learned information in hierarchy during NREM sleep (Fig. 4d). When we counted the quintuple coactivation index between neurons representing the 4 premise pairs and the inference patterns, we found higher synchronized activity during REM 2 than NREM 2 (Fig. 4e).

These results indicate the complementary role played by NREM and REM sleep. The learned knowledge is organized during NREM sleep, while it synchronizes with inferred information during REM sleep that may increase the strength of inference-related patterns leading to inferential behavior. This two-step mechanism may explain the necessity of both NREM and REM sleep for building cognitive inference. Moreover, neurons constituting patterns of the T1 session

did not synchronize with those representing either training or inference sessions (Fig. 4c), supporting that the T1 session is not critical for the inferential behavior observed in T2 and T3.

### REM sleep inspires inference from limited training in the MEC→ACC circuit

To establish a causal link between ACC coactivation code and inference processing, we sought to identify and manipulate a neural circuit that is involved in ACC coactivity formation. ACC coactivity code could rely on synaptic inputs from the connected upstream medial entorhinal cortex (MEC) which is thought to be involved in transitive inference[42,43]. We tested the sufficiency of such coactivation to elicit inferential behavior from inadequate training. Mice were injected into MEC with AAV9 encoding light-sensitive neuronal activator (oChIEF) under the control of human synapsin (hSyn) 1 promoter (Fig. 5a). Mice were exposed to an incomplete/modified protocol of the transitive inference task, in which the randomized training, which is necessary to create inference (Fig. 1d, e), was replaced with optogenetic activation of axonal terminals of the MEC neurons in the ACC at 4 Hz during either NREM or REM sleep (Fig. 5b, c, and Supplementary Fig. 7b). All groups achieved the criterion for correct performance during training of the premises (Fig. 5d). Without both randomized training and MEC→ACC circuit activation, mice did not infer correctly, as demonstrated by a number of correct trials that was not significantly different from chance level in the B and D context tests (Fig. 5e). In contrast, artificial activation of the MEC→ACC network during REM, but not NREM, sleep resulted in inference in T2, T3, and T4, in which mice exhibited significantly more correct trials than those seen in T1, and significantly higher than the chance level; faster selection of the rewarded context was also found (Fig. 5e, f, and Supplementary Fig. 9). These results highlight the dissociable contribution of sleep stages (NREM and REM sleep) in inference processing, and indicates that activating the cortical network during REM sleep is sufficient for the creation of novel conclusions. All groups were proficient during the testing of the non-inference and the premise pairs (Supplementary Fig. 2d).

## Discussion

Our study clarifies the superiority of the subconscious brain over the conscious brain in restructuring existing stores of information before abstracting logical inference. During sleep, multiple related memories were connected to construct a knowledge structure, which facilitated innovative inference, a capacity that was not present shortly after learning. The discrepancy in the performance after awake and sleep manipulations indicates the necessity of offline, but not online, cortical activity to trigger inference. Sleep deprivation and optogenetic inhibition experiments revealed the ability to interfere with the emergence of inference while preserving memories of the original premises, thereby suggesting that stored knowledge and inferential information have unique representations and/or distinct locations in the brain. The artificial induction of inference by initiating communication between the MEC and ACC during REM sleep implies that the dynamic interplay between cortical neural codes for related experiences is critically important for the emergence of inferential behavior.

Our study demonstrates that offline brain activity allows the abstraction of new information from previous experiences, which differs from previous findings showing the online emergence of inferential knowledge during subsequent experiences[37,44]. Furthermore, causal evidence for the link between inference and offline brain activity has been lacking[37,44]. Our study demonstrates the offline and spontaneous emergence of inference without prior priming and provides causal evidence for the necessity and sufficiency of cortical activity during sleep to inspire inferential knowledge. Our demonstration is consistent with previous reports showing the crucial role of sleep in inspiring insight[2,45] and extracting inferential information[3].

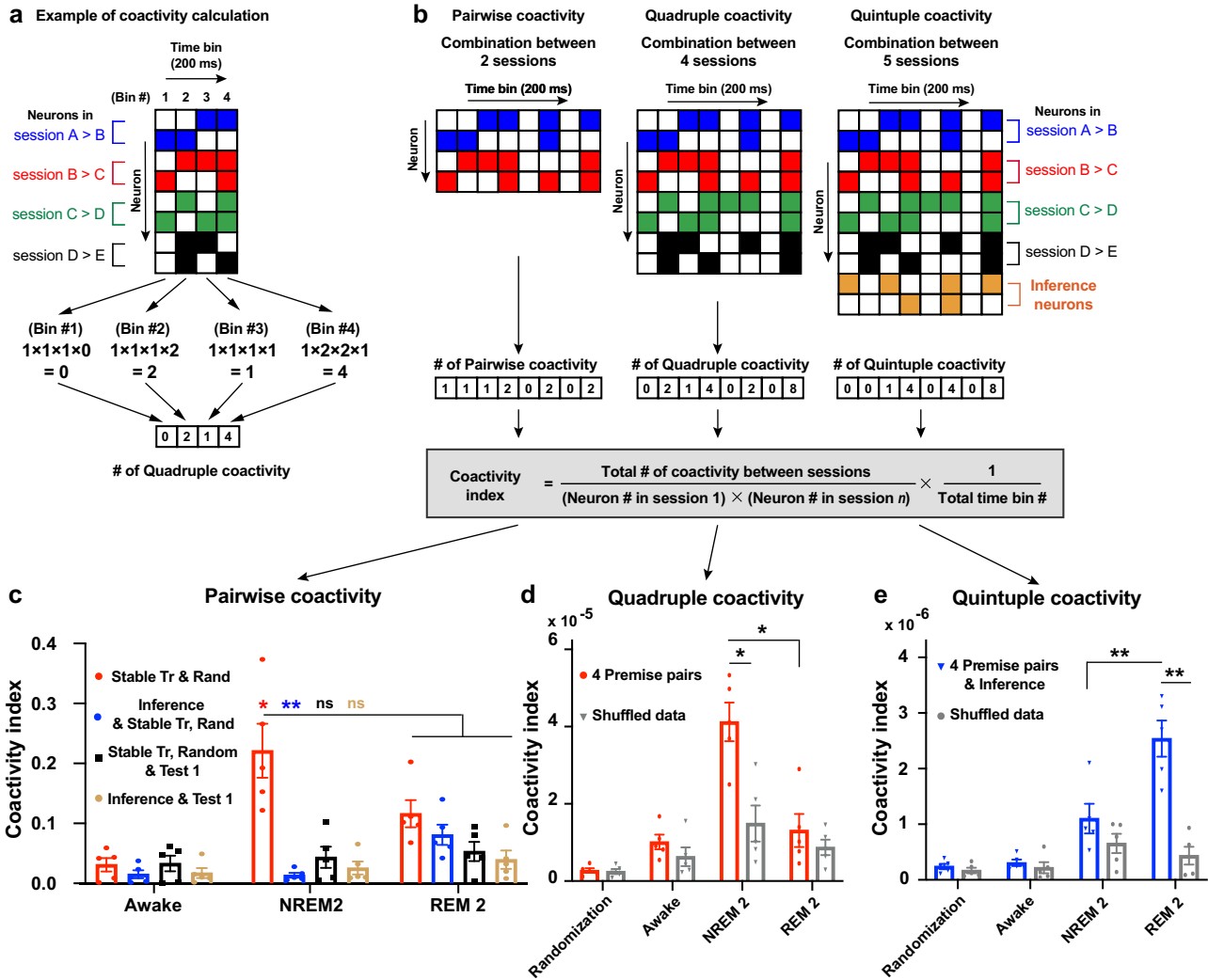

**Fig. 4 | NREM and REM sleep coordinate to build synchronized inference representation with learned knowledge representation. a** Explanation of the coactivity calculation method. **b** Examples of raster plots showing coactivity analysis. This analysis calculates coactivity by normalizing the number of synchronizations every 200 ms among different neuronal subpopulations representing different sessions. Coactivity analysis was done for neurons belonging to different patterns representing 2 sessions (Pairwise coactivity, left), representing 4 sessions (Quadruple coactivity, middle), and representing 5 sessions (Quintuple, right).

Each color denotes neurons representing a specific session. The equation used to calculate the coactivity index (bottom). **c**–**e** Coactivity index between 2 sessions (**c**), 4 sessions (**d**), and 5 sessions (**e**) with the shuffled data; $n = 5$ mice. Statistical comparisons were made using paired $t$-test (**c**–**e**). *$P < 0.05$; **$P < 0.01$; ns not significant ($P > 0.05$). Data are presented as the mean ± standard error of the mean (s.e.m.). Source data are provided as a Source Data file. Detailed statistics are shown in Supplementary Data 1.

A recent study on human subjects has shown that stage 1 NREM sleep might be sufficient to inspire creative problem-solving[45], while our study showed that NREM sleep was necessary, but not sufficient to build inferential knowledge. This discrepancy might be attributed to differences between both tasks, in terms of task modalities and task timeline. Also, there are differences between mouse and human brain that put predictable limitations on relating data across species. These notions may explain the differences between the inference observed in our study and that observed in a previous study on human subjects[3]. Both studies have shown that inference was not developed shortly after learning the premise pairs. However, upon examining the first-order & second-order inference, they found that only second-order inference was sleep-dependent while first-order inference was sleep-independent[3]. Conversely, we proved that first-order inference was sleep-dependent using cellular recording and causal optogenetic manipulation experiments. This finding may suggest that more higher-order inference is even more likely to be sleep-dependent; therefore, we opted not to test the sleep dependence of second-order inference.

On the other hand, non-inference relational memory (A & E) was independent of sleep, likely because A was always rewarded and E was never rewarded. While cellular recording and causal optogenetic manipulation experiments provide valuable insights into the mechanism underlying inferential behavior at the cellular level, these techniques are invasive and cannot be directly applied to human subjects. This limits the translation of the discovered mechanisms to a full understanding of human behavior.

A recent study has reported that prolonged light activation of ArchT on the hippocampus may lead to a rebound increase in hippocampal activity[46]. Therefore, the observed failed inference in Fig. 2 might be due to either a decrease in ACC activity or a rebound in overall ACC activation which might interfere with ACC computations. The latter possibility is less likely to be the reason for failed inference since ACC activation during REM sleep boosted inference emergence (Fig. 5e). On the other hand, previous reports showed that prolonged light stimulation may cause heat-induced neuronal damage which alters cell behavior and neuronal excitability[47,48]. However, we

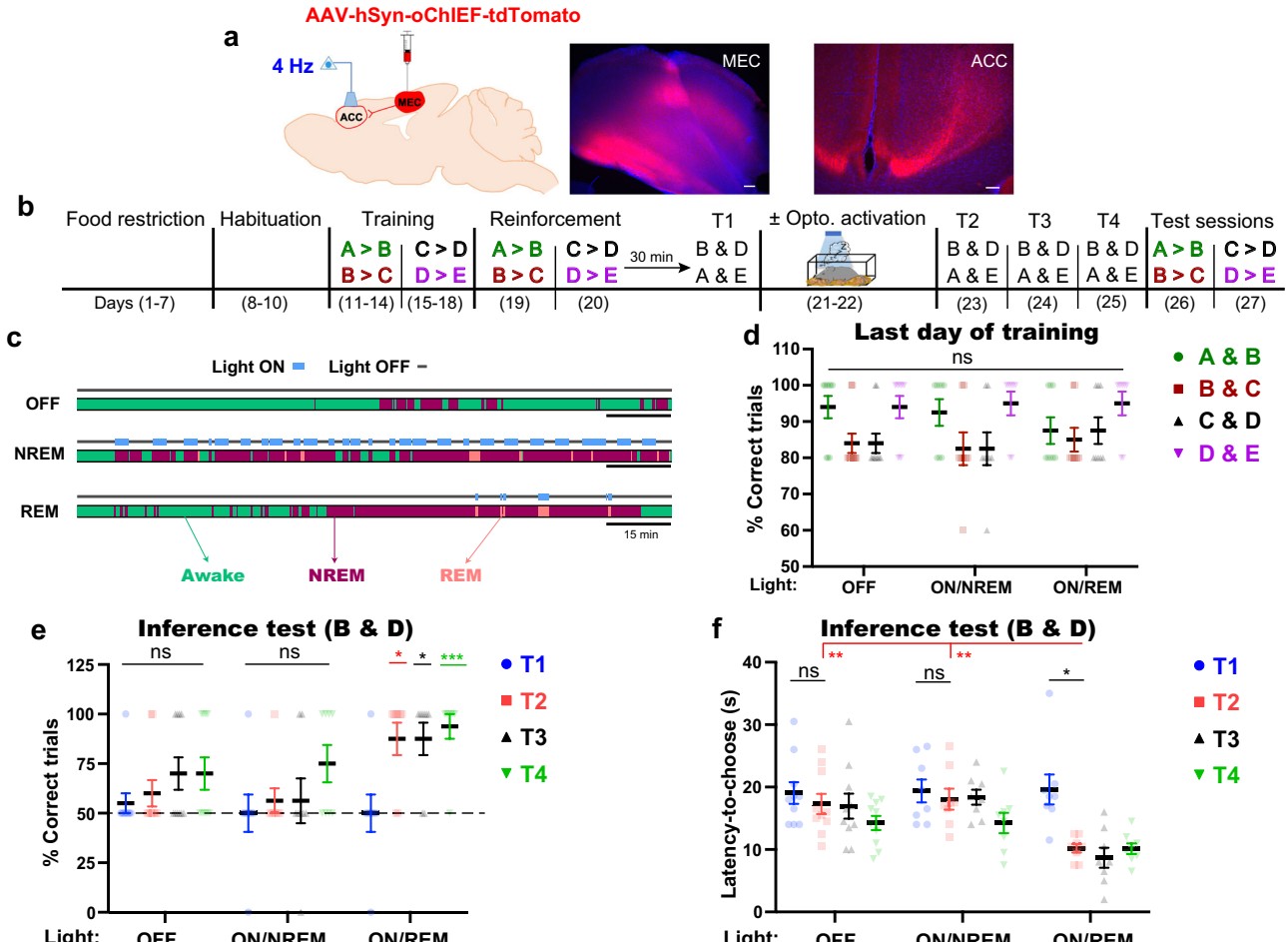

**Fig. 5 | MEC→ACC crosstalk during only REM sleep is sufficient to inspire inference from inadequate training. a** The strategy for labeling MEC neurons and targeting their terminals in the ACC (left), and expression of oChIEF-tdTomato (red) in MEC neurons (middle) and their terminals in the ACC (right). Blue, 4′,6-diamidino-2-phenylindole (DAPI) staining. Scale bars, 100 μm. **b** The behavioral schedule used to manipulate the MEC→ACC circuit during different sleep stages. **c** Diagram showing the sleep stage-specific manipulation. Scale bar, 15 min. **d** Performance during the last day of training (day 14 for A > B and B > C; day 18 for C > D and D > E) for each premise pair. **e, f** Performance during test sessions; the percent of correct trials during the inference test (**e**), the latency time to choose

during the inference test (**f**). T1, test 1; T2, test 2; T3, test 3; T4, test 4. *n* = 10 mice for the light-off group; *n* = 8 mice for the non-rapid eye movement (NREM) sleep group; *n* = 8 mice for the rapid eye movement (REM) sleep group. Statistical comparisons were made using a two-way repeated-measures analysis of variance (ANOVA) with Tukey's multiple comparison test (**d**–**f**). *$P < 0.05$; **$P < 0.01$; ***$P < 0.001$; ns not significant ($P > 0.05$). Data are presented as the mean ± standard error of the mean (s.e.m.). The Experiment was independently repeated six times. Source data are provided as a Source Data file. Detailed statistics are shown in Supplementary Data 1.

confirmed behaviorally that the optogenetic protocol used in this study did not cause brain damage since both the learned and inferred knowledge were not affected by prolonged light stimulation to ACC during awake state. Also, mouse performance during testing the premise pairs after long NREM inhibition was comparable to that observed during the training stage (Fig. 2d & Supplementary Fig. 2c).

In Fig. 1, when mice made the correct choice, they were not rewarded to exclude the possibility of direct learning of B > D during test sessions. However, in the optogenetic inhibition experiment (Fig. 2), mice were rewarded in context B during test sessions to confirm that the observed failed inference was due to manipulating the offline ACC computations, not due to lack of motivation from non-rewarding after a correct choice. Providing rewards during test sessions did not overcome the effect of manipulation, which strengthen our conclusion. Since test sessions consist of only 2 trials and the percentage of correct trials (rewarded) in case of failed inference is 0% or 50%, therefore mice would receive only 1 reward pellet per day that is not sufficient for mice to prefer context B over D. The successful inference in the absence of reward (Fig. 1) along with failed inference in the presence of reward (Fig. 2) during test session indicates that

reward during test sessions is not necessary for the inferential behavior. These notions can further explain the results that the inferential behavior (B > D) computed from arranging the randomized pairs in the hierarchy (Fig. 1d, e) was more evident than the behavior resulting from direct rewarding context B over D (Fig. 2 & Supplementary Fig. 5).

REM sleep is characterized by vivid dreams[25] and could be initiated by activating specific dopaminergic neurons in the amygdala[49]. Prior investigations into the role of REM sleep in cognition have found that it is necessary for consolidating both contextual and emotional memories which involve amygdala-related networks[8,50]. Furthermore, REM sleep has been proposed to be involved in creative problem-solving by enhancing the formation of associative networks[22,25]. Here, we present evidence of a causal link between REM sleep and creativity using loss and gain of function experiments. Activating the MEC→ACC network at 4 Hz during REM sleep may provide an ideal setting for dreaming the inference, thus leading to innovative decisions. This suggestion is consistent with a previous study demonstrating that 4 Hz oscillations, dominant in PFC, were phase coupled with theta rhythm, leading to modulation of neuronal spikes in the PFC; hence supporting the processing of information[51]. Since theta rhythm (4–8 Hz) and

network oscillations support transient synchronization between brain regions[51], we hypothesize that artificial stimulation of the MEC→ACC network at 4 Hz may mimic the physiological circuit mechanism of inference emergence.

The dissociable roles of REM and NREM sleep in processing the inference suggest that there are distinct offline mechanisms underlying the formation of knowledge networks. Since NREM sleep precedes REM sleep, we propose that NREM sleep prepares the knowledge network for REM-induced restructuring and the creation of novel associations[25]. This may explain why NREM sleep is crucial, but not sufficient, for the emergence of inference. The greater contribution of REM sleep over NREM sleep in inference development supports earlier findings that performance in anagram word puzzles is better after awakening from REM sleep than after awakening from NREM sleep[52]. Our Ca$^{2+}$ imaging data suggests that NREM sleep may connect and organize the learned information in a hierarchy, while REM sleep may build and stabilize the inference representation by creating new connections with the learned knowledge through co-reactivating their representations (Supplementary Fig. 10).

Ca$^{2+}$ imaging unveiled a coactivity coding logic during the offline state, which may link temporally segregated events and organize them in hierarchical order, allowing for building new information for future use. This demonstration is consistent with a previous study showing that synchronized offline reactivations are necessary for assimilating two subtly related memories and eventually revealing the implicit commonality between them[41]. These data indicate that offline coactivation coding could underlie sleep-triggered creativity.

The capacity of REM sleep to facilitate inference from inadequate knowledge could inspire the development of novel approaches to boost cognitive performance. A recent study has reported that dopamine signaling in the amygdala is crucial for the transition from NREM to REM sleep[49]. Taken together, our study shed light on the mechanism of intellectual and cognitive impairments that occur in diseases that involve abnormalities in dopamine signaling like Parkinson's disease and Attention-deficit hyperactivity disorder. The ability of REM sleep to flexibly restructure our knowledge could offer new behavioral repertoires to facilitate higher-order brain functions, such as implicit learning, decision-making, and creative thinking.

## Methods

### Animals
Naïve wild-type male C57BL/6J mice were purchased from Sankyo Labo Service Co. Inc. (Tokyo, Japan) and maintained on a 12 h light/dark cycle at a controlled temperature (24 °C ± 3 °C) and humidity (55% ± 5%) with free access to food and water. Mice used in behavioral experiments were 14–20 weeks old. All experimental procedures with animals were congruent with the guidelines of the National Institutes of Health. The Animal Care and Use Committee of the University of Toyama approved all animal experiments with approval number (A2022MED-9).

### Viral constructs
For the optogenetic inhibition experiment (Fig. 2 and Supplementary Fig. 6), the AAV viral vector AAV-CaMKII-ArchT-eYFP (3.15 × 10$^{13}$ vg/mL) was used. pAAV-CaMKII-ArchT-eYFP was kindly donated by Dr. K. Deisseroth. The recombinant AAV9 production was performed using a minimal purification method, and viral genomic titer was subsequently calculated as described previously[53]. Briefly, the pAAV recombinant vector was produced using HEK293 T cells (AAV293; 240073, Agilent Tech, CA, USA) cultured in 15 cm dishes (Corning, NY, USA). Cultured cells were maintained in Dulbecco's Modified Eagle Medium (D-MEM) (11995-065, GIBCO life technologies, USA) supplemented with 10% fetal bovine serum (FBS) (10270106, GIBCO life technologies, USA), 1% 2 mM L-Glutamine (25030-149, GIBCO Life Technologies, USA), 1% 10 mM non-essential amino acid (MEM NEAA 100×, 11140-050, GIBCO

Life Technologies, USA), and 1% (100×) penicillin-streptomycin solution (15140-148, GIBCO Life Technologies, USA). Confluent (70%) HEK293 T cells were transfected using a medium containing the constructed expression vector, pRep/Cap, and pHelper (240071, Agilent Technologies, Santa Clara, CA, USA) mixed with the transfection reagent polyethyleneimine hydrochloride (PEI Max, 24765-1, Polysciences, Inc., Warrington, PA, USA) at a 1:2 ratio (W/V). After 24 h, the transfection medium was discarded, and cells were incubated for another 5 days in an FBS-free maintenance medium. On day 6, the AAV-containing medium was collected and purified from cell debris using a 0.45 µm Millex-HV syringe filter (SLHV033RS, Merck Millipore, Germany). The filtered medium was concentrated and diluted with D-PBS (14190-144, GIBCO Life Technologies, USA) twice using the Vivaspin 20 column (VS2041, Sartorius, Germany) after blocking the column membrane with 1% bovine serum albumin (01862-87, Nacalai Tesque, Inc., Japan) in PBS. To further calculate the titer, degradation of any residual cDNA in the viral solution from production was first assured by benzonase nuclease treatment (70746, Merck Millipore, Germany). Subsequently, viral genomic DNA was obtained after digestion with proteinase K (162-22751, FUJIFILM Wako Pure Chemical, Osaka, Japan), extraction with phenol/chloroform/isoamyl alcohol 25:24:1 v/v, then precipitation with isopropanol and final dissolution in TE buffer (10 mM Tris [pH 8.0], 1 mM EDTA). Titer quantification for each viral solution, referenced to that of the corresponding expression plasmid, was done by real-time quantitative PCR (qPCR) using THUDERBIRD SYBR qPCR Master Mix (QRS-201, Toyobo Co., Ltd, Japan) with the primers 5′-GGAACCCCTAGTGATGGAGTT-3′ and 5′-CGGCCTCAGT-GAGCGA-3′ targeting the inverted terminal repeat (ITR) sequence. The cycling parameters were adjusted as follows: initial denaturation at 95 °C for 60 s, followed by 40 cycles of 95 °C for 15 s and 60 °C for 30 s.

For the in vivo Ca$^{2+}$ imaging experiment (Figs. 3, 4), the AAV viral vector AAV-Syn-janelia-GCaMP7b (8.36 × 10$^{13}$ vg/mL) was used. The pAAV-Syn-janelia-GCaMP7b[54] was purchased from Addgene (Cambridge, MA, Plasmid #104489). The recombinant AAV9 production was performed using a minimal purification method, and viral genomic titer was subsequently calculated as described above and previously[53].

For the optogenetic activation experiment (Fig. 5), the AAV viral vector AAV-hSyn1-oChIEF-tdTomato (2.1 × 10$^{13}$ vg/mL) was used. pAAV-hSyn1-oChIEF-tdTomato was purchased from Addgene (Cambridge, MA, Plasmid #50977). Recombinant AAV vectors were produced by transient transfection of HEK293 cells with the vector plasmid, an AAV3 rep and AAV9 vp expression plasmid, and an adenoviral helper plasmid, pHelper (Agilent Technologies, Santa Clara, CA), as previously described[55,56].

**Surgery.** Mice (10–14 weeks old) were given an intraperitoneal anesthesia injection containing 0.75 mg/kg medetomidine (Domitor; Nippon Zenyaku Kogyo Co., Ltd., Japan), 4.0 mg/kg midazolam (Fuji Pharma Co., Ltd., Japan), and 5.0 mg/kg butorphanol (Vetorphale, Meiji Seika Pharma Co., Ltd., Japan) before being placed, when sedated, on a stereotactic apparatus (Narishige, Tokyo, Japan). After surgery, an intramuscular injection of 1.5 mg/kg atipamezole (Antisedan; Nippon Zenyaku Kogyo Co., Japan), an antagonist of medetomidine, was administered to boost recovery from sedation. Mice were home-caged for 3 weeks to recover from surgery before initiating behavioral experiments. All virus injections were done using a 10 µL Hamilton syringe (80030, Hamilton, USA) that was fitted with a mineral oil-filled glass needle and wired to an automated motorized microinjector IMS-20 (Narishige, Japan).

For the optogenetic inhibition experiments (Fig. 2 and Supplementary Fig. 6), 500 nL of AAV viral vector were injected at 100 nL min−1 bilaterally into either the ACC (from bregma: +1.0 mm anteroposterior [AP], ±0.35 mm mediolateral [ML]; from the skull surface: +1.5 mm dorsoventral [DV]) or prelimbic cortex (PL) (from bregma: +2.0 mm AP, ±0.35 mm ML; from the skull surface: +1.8 mm DV). The

glass injection tip was maintained after injection at the target coordinates for an additional 5 min before being removed. Then, a double-guide cannula (C2002GS-5-0.7/SPC, diameter 0.29 mm, Plastics One Inc., USA) composed of two 0.7 mm-spaced stainless-steel pipes protruding for 2 mm from the plastic cannula body was bilaterally inserted either 1.0 mm ventral to the skull surface at the ACC coordinates or 1.3 mm ventral to the skull surface at the PL coordinates. Guide cannulas were fixed using dental cement (Provinice, Shofu Inc., Japan) that was used to fix micro-screws that were anchored into the skull near bregma and lambda. After complete fixation, a dummy cannula (C2002DCS-5-0.7/SPC, protrusion 0 mm, Plastics One Inc., USA) was attached to the guide cannula to protect it from particulate matter.

In parallel, a custom-built electroencephalogram/electromyography (EEG/EMG) 5-pin system was installed into the skull as previously described[39]. Briefly, electrodes were screwed into the skull over the parietal cortex for EEG recording, over the right cerebellar cortex as a ground, and over the left cerebellar cortex as a reference. Additionally, two wires were implanted into the neck muscle for EMG recording. Finally, dental cement was used to fix all system screws in place.

For the optogenetic activation experiment (Fig. 5), 500 nL of the AAV viral vector was injected at 100 nL min−1 bilaterally into the MEC (from bregma: +4.8 mm AP, ±3.3 mm ML, +3.3 mm DV). The remaining procedure was as described earlier in the optogenetic inhibition section.

For the Ca²⁺ imaging experiment (Figs. 3, 4), 500 nL of AAV₉-CaMKII::GCaMP7 was injected at 100 nL min⁻¹ unilaterally into the left ACC (+1.0 mm AP, +0.35 mm ML, +1.5 mm DV). After 2 weeks of recovery from AAV injection surgery, re-anesthetized mice were placed once again on a stereotactic apparatus to implant a gradient index (GRIN) lens[6,41,57] (0.5 mm diameter, 4 mm length; Inscopix Inc., USA) into the center of injection (from the skull surface: +1.2 mm DV) using custom-made forceps attached to a manipulator (Narishige, Japan). A low-temperature cautery was used to emulsify bone wax into the gaps between the GRIN lens and the skull, and then the lens was anchored in place using dental cement. Additionally, a custom-built EEG/EMG 5-pin system was installed and cemented into the skull, as mentioned earlier. Three weeks after GRIN implantation, mice were re-anesthetized and placed back onto the stereotactic apparatus to set a baseplate (Inscopix Inc.), as described previously[41,57]. In brief, a Gripper (Inscopix Inc.) holding a baseplate attached to a miniature microscope (nVistaHD v3; Inscopix, Inc.) was lowered over the previously set GRIN lens until visualization of clear vasculature was possible, indicating the optimum focal plane. Dental cement was then applied to fix the baseplate in position to preserve the optimal focal plane. Mice recovered from surgery in their home cages for 1 week before behavioral imaging experiments began.

### Transitive inference task

The behavioral experiments were done in a soundproof room. The arena consists of 5 different contexts (A, B, C, D, and E; Fig. 1a). Context A was a gray triangle (200 mm length × 300 mm height) with a smooth, gray acrylic floor and walls without any patterns. Context B was a gray square (200 mm length × 200 mm width × 300 mm height) with a green plastic floor with a pointy texture and walls without any patterns. Context C was a blue square (200 mm length × 200 mm width × 300 mm height) with a smooth, blue acrylic floor and walls without patterns. Context D was a transparent square (200 mm length × 200 mm width × 300 mm height) with a spongy, orange floor and walls that were covered with a characteristic pattern (black circles on a white background). Context E was a transparent triangle (200 mm length × 300 mm height) with a smooth, transparent acrylic floor and walls that were covered with a characteristic pattern (black vertical lines on a white background). The arena was differently positioned in the room

across sessions to avoid the effect of spatial distal cues on mice performance (Supplementary Movies 1, 2).

**Food restriction.** Mice were kept under a food restriction protocol during the task. Mice were provided with one 3 g food pellet and one 0.05 g sucrose tablet per day until mice reach 80–85% of their original weight. The sucrose tablets were put in a small Petri dish inside the home cage during the first two stages (food restriction and arena habituation); the dish was removed after starting the training stage and was never put it back, which served to teach mice that they would no longer receive any sucrose tablets in the home cage and the only way to receive the reward in the contexts was to perform the task correctly.

**Arena habituation.** On the first day, mice were put in each context separately for 5 min. The contexts were closed to force mice to explore each context for the entire 5 min. The exposure to the contexts was random across mice (for example, one mouse was placed in A then C then D then B then E, and another mouse was placed in C then B then E then A then D). On the second and third days, all contexts were open and mice were put in the start point for free exploration of the whole arena for 15 min.

**Training.** The 14-day training phase was divided into 4 consecutive days for the first two pairs, 4 days for the other two pairs, 2 days reinforcement, and 4 days randomization. Every day, mice were exposed to two sessions, and each session consisted of five trials for a premise pair without inter-trial intervals. The four premise pairs were A > B, B > C, C > D, and D > E. Therefore, the hierarchy was A > B > C > D > E. For the incomplete hierarchy experiment (Supplementary Fig. 5), the four premise pairs were A > B, B > C, E > D, and D > C. Therefore, the hierarchies were A > B > C and E > D > C. Premise pair training was considered successful when mice reached 80% correctness.

In the first trial of every session, mice were put at the starting point and allowed to explore the arena and enter the two contexts to identify the learned pair. After entering both contexts once, if they entered the correct context again and remained there for more than 10 s, they received the sucrose tablet, and the trial ended after the tablet had been consumed. If they stayed in the non-rewarded context (wrong choice) for more than 10 s, the trial ended, and in the next correct trial, they received the sucrose tablet after 20 s (rather than 10 s). After each trial, mice were directly returned to the starting point to start the next trial without any inter-trial interval. After completing all trials, mice were returned to their home cage for 30 min before starting the next session. After finishing two sessions, the mice returned to their home cages and received a 2-g food pellet.

For randomization during training, mice were exposed to new combinations of premise pairs every day for 4 days. These combinations were different from those presented during the initial training and reinforcement. For example, (A & B) pair was co-presented with (B & C) pair on the same day during the initial training and reinforcement, therefore during the randomization stage, (A & B) pair was co-presented with (C & D) and (D & E) pairs on two different days; and the same was applied on the other pairs to ensure the complete interaction between all premise pairs. The order of presenting the premise pairs is different across animals within the same group (Table 1). This is different from training with reinforcement, in which the A & B pair is usually presented with the B & C pair, and the C & D pair is usually presented with the D & E pair. For the optogenetic activation experiment (Fig. 5 and Supplementary Fig. 9), training was done for 10 days only without the 4 days of randomization.

**Testing.** After the training phase, mice were exposed to two types of testing for 4 consecutive days. The inference test utilized a novel pair (B & D), while the non-inference test utilized another novel pair (A & E).

**Table 1 | Example of the order of presenting the premise pairs during the randomization stage**

| Randomization Day # | 1 | 2 | 3 | 4 |
|---|---|---|---|---|
| Day # in the protocol | Day # 11 | Day # 12 | Day # 13 | Day # 14 |
| Mouse # 1 | A > B & C > D | B > C & C > D | A > B & D > E | B > C & D > E |
| Mouse # 2 | A > B & D > E | A > B & C > D | B > C & D > E | B > C & C > D |
| Mouse # 3 | B > C & C > D | B > C & D > E | A > B & D > E | A > B & C > D |

Test 1 was done on the last day of training, while tests 2, 3, and 4 were done on the following 3 days.

Every day, mice were exposed to two sessions, each of which consisted of two trials for each test without any inter-trial intervals. After completing all trials, mice were returned to the home cage for 30 min before starting the next session. The protocol of test sessions was the same as for the training sessions. When they made the correct choice, mice were not rewarded in the inference group (Fig. 1 & Supplementary Fig. 4) and in both awake and sleep groups (Supplementary Fig. 6) to exclude the possibility of direct learning. However, in the remaining experiments, when mice made the correct choice, they were rewarded with a sucrose tablet in context B (Figs. 2, 3, 5 & Supplementary Figs. 3, 5, 9). In the latter figures (control & manipulation experiments), we provided rewards to mice to keep their motivation and to confirm that any failed performance was due to the manipulation, not due to lack of motivation from non-rewarding after the correct choice. Both the B and D contexts were square and located at the end of the arena (the same distance from the starting point) to avoid any preference that was due to geometry and distance from the starting point. Performance during the B and D test was also not due to right or left training preferences, because the design of the premise pairs was based on an equal distribution of right and left preferences (the correct choice was located to the left in two pairs and to the right in the other two pairs). After completing the inference and non-inference tests, mice were tested with the original premise pairs to confirm that the performance during the inference test was due to remembering the original memories. The arena was cleaned using water and 70% ethanol after each subject. Sleep deprivation (Fig. 1) was done for 4 h by gentle touching of the home cage after test sessions (after T1, T2, T3, and T4).

For the optogenetic inhibition experiments (Fig. 2 and Supplementary Fig. 6), optogenetic inhibition of the ACC or PL was induced during wakefulness or sleep in a 4 h session. On each testing day, immediately after the test sessions, mice were anesthetized using isoflurane, and two branch-type optical fibers (internal diameter, 0.25 mm) were inserted and fitted into their housing with a cap, which anchored the inserted optical fibers with screws around the guide cannula. The tip of the optical fiber protruded 0.2 mm below the guide cannula (ACC, DV 1.2 mm from the skull surface; PL, DV 1.5 mm from the skull surface). Mice attached to the optic fibers were then placed in a sleep box and simultaneously connected to an EEG/EMG recording unit and an optical swivel wired to a laser unit (9–12 mW, 589 nm). The delivery of continuous light was manually controlled. After the session, mice were detached from the EEG/EMG recording and light delivery systems, and the fibers were removed from the cannula under anesthesia.

For the in vivo Ca$^{2+}$ imaging experiment (Figs. 3, 4), attachment and removal of the microendoscope was performed under 3% isoflurane anesthesia. Mice attached to the microendoscope were returned to their home cages to recover for at least 30 min before and after the behavioral session. Three days before context habituation sessions, mice were habituated to the microendoscope attachment for 10 min per day in their home cages. Calcium imaging data acquisition started from the last day of context habituation. Mice were exposed to the above-mentioned behavioral protocol with slight modifications in the initial training stage. Mice were exposed to two particular premise

pairs for 2 successive days, then they encountered a single premise pair per day on the subsequent 2 days to collect the corresponding Ca$^{2+}$ data of each premise pair on a separate day. Offline Ca$^{2+}$ imaging data was collected from the sleep period (NREM 1 & REM 1) after context habituation, the awake session after Test 1 (Day 24), and the sleep period (NREM 2 & REM 2) after Test 1. Ca$^{2+}$ imaging data was collected from the first 2 min of NREM sleep (single NREM epoch that lasts for more than 2 min) and from the first 2 min of REM sleep (multiple REM epochs; number of epochs across animals is 3 ± 1).

For the optogenetic activation experiment (Fig. 5), optogenetic activation to MEC terminals in the ACC was performed during NREM or REM sleep for two consecutive days. One day after test 1, mice were anesthetized using isoflurane, and two branch-type optical fibers (internal diameter, 0.25 mm) were inserted and fitted into their housing with a cap, which anchored the inserted optical fibers by screws around the guide cannula. The tip of the optical fiber protruded 0.2 mm below the guide cannula (DV 1.2 mm from the skull surface). Mice attached to the optic fibers were then placed in a sleep box, and simultaneously connected to an EEG/EMG recording unit and an optical swivel wired to a laser unit (9–12 mW, 473 nm). The delivery of laser stimulation (4 Hz, 15 ms pulse width) was manually controlled using a schedule stimulator in time-lapse mode (Master-8 pulse stimulator, A.M.P.I.). In REM-stimulated mice, 4 Hz light stimulation was delivered during all REM sleep that occurred within the 4 h sleep session; the total duration of light stimulation was less than 7 min per mouse. For NREM-stimulated mice, light stimulation was delivered for a maximum of 3 min per epoch with a 3 min inter-epoch interval. After the session, the mice were detached from the EEG/EMG recording and light delivery systems, and the fibers were removed from the cannula under anesthesia.

For the optogenetic activation experiment (Supplementary Fig. 9), the procedure was the same as that described in the previous paragraph, related to Fig. 5, with a modification in the stimulation protocol; the total duration of light stimulation was modified to be less than 7 min per mouse, with a maximum of 3 min per epoch. This modification was done to avoid prolonged light stimulation and to mimic the protocol used in REM-stimulated animals.

### Behavioral analysis

All sessions were captured with an overhead web camera (Logitech HD pro C920) mounted on a vertical stand. The time spent in each context during the habituation phase was counted. During training and testing, the trial was considered to be correct if mice spent >10 s in the assigned context. The percent of correct trials during training and testing sessions was calculated as follows: % correct trials = (number of correct trials / total number of trials) * 100. We set 80% correct trials during the training phase as the criterion for successful learning of the premise pairs.

### In vivo Ca$^{2+}$ imaging data acquisition and analyses

Ca$^{2+}$ signals produced from GCaMP7 protein were captured at 20 Hz with nVista acquisition software (Inscopix, CA, USA) at the maximum gain and optimal power of LED of nVistaHD. Ca$^{2+}$ imaging movie recordings of all behavioral sessions were then extracted from the nVista Data acquisition (DAQ) box (Inscopix, CA, USA). Using Inscopix data processing software (v1.9.4.3801, Inscopix, USA), movies were temporally stitched together to create a full movie that contained recordings of all sessions across days, which were spatially downsampled (2×), and then corrected for across-session motion artifacts against a reference frame that was chosen from any session that had a clear landmark "vasculature". Further motion correction was then applied using Inscopix Mosaic software (v1.2.0, Inscopix, USA), as previously described[6]. The full movie was then temporally divided into individual sessions using Inscopix Mosaic software. Each movie of individual sessions was then low bandpass filtered using Fiji software

(version 1.53q, NIH, USA) to reduce noise, as described previously[6]. The fluorescence signal intensity change ($\Delta F/F$) for each session was subsequently calculated using Inscopix Mosaic software according to the formula $\Delta F/F = (F-Fm)/Fm$, where F represents each frame's fluorescence and Fm is the mean fluorescence for the whole session's movie. Afterward, movies representing each session were re-concatenated again to generate the full movie for all sessions in the $\Delta F/F$ format. Finally, cells were identified using an automatic sorting system, HOTARU (version 3.3.2), and each cell's $Ca^{2+}$ signals over time were extracted in a ($Ď$; time × neuron) matrix format, as previously described[6]. Further processing was performed using codes written in MATLAB 2020b (Mathworks, USA) to remove the low-frequency fluctuation and background noise by subjecting output $Ca^{2+}$ signals to high-pass filtering with a 0.01 Hz threshold, and to then calculate z-scores from the mean of each session, whereby negative values were replaced with zero. $Ca^{2+}$ events were finally extracted after cutting off signals below 3 SD from the local maxima of the $\Delta F/F$ signal of each session.

## Neural ensembles analyses

Neural ensembles representing a range of neurons co-firing together were calculated using an unsupervised statistical method using PCA-ICA analysis as previously described[58,59]. Initially, frames were binned to 0.2 s, then to prevent the bias due to differences in the average firing rates, neuronal $Ca^{2+}$ transients were normalized for each neuron by a z-score transform:

$$Z_{i,b} = \frac{x_{i,b} - \mu_{x_i}}{\sigma_{x_i}}$$

where for neuron $i$, $z_{i,b}$ is its z-scored $Ca^{2+}$ transients in bin $b$, $x_{i,b}$ its $Ca^{2+}$ transients in bin $b$, $\mu_{x_i}$ its mean $Ca^{2+}$ transients across all bins and $\sigma_{x_i}$ the s.d. of its $Ca^{2+}$ transients. The calculation of the principal components and the determination of the number of significant components were carried out according to previously presented methods.

To quantify the extent to which ACC neurons contribute to the formation of neuronal ensembles, we applied reconstruction ICA to significant principal components as described previously[58,59]. Ensemble activation strength was then calculated through:

$$R_k(t) = Z(t)^T P_k Z(t)$$

where $R_k(t)$, the activation strength of the ensemble $k$ at the time $t$.

Wilcoxon rank-sum test was then used to reveal which of the identified ensembles are significantly different between reference and target sessions. $R_k(t) > 2$ were set as a threshold for ensembles to be included in the test.

## Coactivity index analysis

The coactivity index was calculated as described previously[57]. The z-scores of neuronal $Ca^{2+}$ events were calculated as described above. The z-scores were binarized by thresholding (>3 Standard Deviations from the $\Delta F/F$ signal) at the local maxima of the $\Delta F/F$ signal and then were temporally down-sampled from 20 to 5 Hz data (200 ms binning). Subsequently, the coactivation index consisting of the number of synchronized activities among neurons -constituting different patterns from different sessions- in each 200 ms time window, was calculated and normalized in each behavioral session. The following equations are used for the calculations of pairwise, quadruple, and quintuple coactivity index:

*Pairwise Coactivity*

$$= \frac{1}{T} \frac{\sum_{t=1}^{T} n_{SessionA}(t).n_{SessionB}(t)}{N_{SessionA}.N_{SessionB}} \qquad (1)$$

*Quadruple Coactivity*

$$= \frac{1}{T} \frac{\sum_{t=1}^{T} n_{SessionA}(t).n_{SessionB}(t).n_{SessionC}(t).n_{SessionD}(t)}{N_{SessionA}.N_{SessionB}.N_{SessionC}.N_{SessionD}} \qquad (2)$$

*Quintuple Coativity*

$$= \frac{1}{T} \frac{\sum_{t=1}^{T} n_{SessionA}(t).n_{SessionB}(t).n_{SessionC}(t).n_{SessionD}(t).n_{SessionE}(t)}{N_{SessionA}.N_{SessionB}.N_{SessionC}.N_{SessionD}.N_{SessionE}} \qquad (3)$$

Where A, B, C, D & E are different behavioral sessions; $n_{SessionA}(t)$ is the number of cells, constituting significant patterns in Session A, that were active in the time bin t; $N_{SessionA}$ is the total number of Session A cells; and T is the total number of time bins for each session.

To exclude the possibility that the observed coactivity results from the difference in activities among cells and sessions, we generated shuffled data (1000 surrogate values) by using the "circshift function of MATLAB"[60].

## Sleep detection data acquisition and online state detection

All EEG/EMG recordings were performed using OpenEx Software Suite v2.32 (RX8-2, Tucker Davis Technologies, USA), as previously described[39] with minor modifications. Briefly, EEG signals were amplified and filtered at 1–40 Hz, while 65–150 Hz was used for EMG; signals were then digitized at a sampling rate of 508.6 Hz. Sleep stages were differentiated using an algorithm file that enabled the calculation and analysis of the EMG root mean square value (RMS), EEG delta power (1–4 Hz) RMS, and EEG theta power (6–9 Hz) RMS. The EMG RMS threshold was optimized according to each subject. Mice were judged to be awake when the EMG RMS exceeded the set threshold value and remained unchanged for three successive 3 s checkpoints. However, when EMG RMS was lower than the threshold, sleep stage differentiation was concluded based on the delta/theta (d/t) ratio value. Briefly, if the d/t ratio exceeded 1 for the consecutive 9 s checking period, the stage was classified as NREM sleep; conversely, it was classified as REM sleep when it was less than 1 for a consecutive 9 s. The state classified by the program was also confirmed by the experimenter through visual inspection of mouse activity and EEG delta-dominant (0.5–4 Hz) or theta-dominant (4–9 Hz) waveforms. EEG/EMG traces recorded during sleep sessions were then extracted using MATLAB codes.

## Histology

After the optogenetic manipulation experiments, mice were deeply anesthetized with 1 mL of combination anesthesia and perfused transcardially with PBS (pH 7.4) followed by 4% paraformaldehyde in PBS. The brains were extracted, then further immersed in paraformaldehyde in PBS for 12–18 h at 4 °C. Subsequently, fixed brains were mixed with 25% sucrose in PBS for 36–48 h at 4 °C before final storage at −80 °C. To obtain coronal sections, brains were sliced into 50 μm sections using a cryostat (Leica CM3050, Leica Biosystems) and were then washed in PBS-containing 12-well culture plates (Corning, NY, USA). The sections were further incubated at room temperature for 1 h with a blocking buffer (3% normal donkey serum; S30, Chemicon by EMD Millipore, Billerica, MA, USA) in PBS solution containing 0.2% Triton X-100 and 0.05% Tween 20 (PBST). After incubation, the buffer was discarded and rat anti-GFP (04404-84, GF090R, Nacalai Tesque Inc., Japan) primary antibody (1:500) in blocking solution was added for further incubation at 4 °C for 24–36 h. At the end of the incubation period, the primary antibody was removed and sections were washed with 0.2% PBST three times for 10 min each. After washing, sections were treated with a complementary secondary antibody, (1:1000) donkey anti-rat IgG Alexa Fluor 488 (A21208, Molecular Probes) in blocking buffer solution at room temperature for 2–3 h. Finally, the sections were treated with DAPI (1 μg/mL, Roche

Diagnostics, 10236276001) for nuclear staining and washed with 0.2% PBST three times for 10 min each before being mounted onto glass slides with ProLong Gold antifade reagent (Invitrogen).

## Confocal microscopy
Images were acquired using a confocal microscope (Zeiss LSM 780, Carl Zeiss, Jena, Germany) with 20× Plan-Apochromat objective lens. All parameters, such as the photomultiplier tube assignments, pinhole sizes, and contrast values, were standardized within each magnification and experimental condition.

## Statistics
Statistical analyses were performed using Prism 8 (GraphPad Software, San Diego, CA, USA). Multiple-group comparisons were performed using an ANOVA with post hoc tests, as shown in the corresponding Fig. legends. Quantitative data are presented as the mean ± SEM.

## Reporting summary
Further information on research design is available in the Nature Portfolio Reporting Summary linked to this article.

## Data availability
All data and resources that supported the findings of this study are available upon request. The datasets supporting this study will be deposited to a public repository when the ongoing studies using the same dataset are published. Source data are provided with this paper.

## Code availability
All codes, used in the manuscript, are available at https://github.com/IdlingBrainUT/Abdou2024_NatureCommunications.

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

## Acknowledgements

We thank Khaled Ghandour and Ali Choucry (University of Toyama) for their valuable discussions and suggestions. We thank all members of the Inokuchi Laboratory for their discussions and support. We thank Mika Ito and Naomi Takino (Jichi Medical University) for their help with the production of the AAV vectors. Also, we thank Noriaki Ohkawa (Dokyyo University) for his help in providing the materials used to build the behavioral arena. We would like to thank Karl Deisseroth (Stanford University) for providing us with pAAV-CaMKII-ArchT 3.0-EYFP. We also thank Ayumu Konno and Hirokazu Hirai (Gunma University) for providing us with the virus preparation protocol. This work was supported by the JSPS KAKENHI (grant number JP18H05213, JP23H05476), the Core Research for Evolutional Science and Technology (CREST) program (JPMJCR13W1, JPMJCR23N2) of the Japan Science and Technology Agency (JST), a Grant-in-Aid for Scientific Research on Innovative Areas "Memory dynamism" (JP25115002) from MEXT, and the Takeda Science Foundation to K.I.; JSPS KAKENHI (grant number JP19K16892) to K.A.; and the Uehara Memorial Foundation scholarship to M.A.; JST SPRING scholarship (grant number JPMJSP2145) to A.I.; Grant-in-Aid for AMED under grant number JP23gm6510028, JSPS KAKENHI Scientific Research(B) (20H03554, 23H02785), the Takeda Science Foundation, the Tamura Science and Technology Foundation to M.N.

## Author contributions

K.A. and K.I. designed the experiments and wrote the manuscript. K.A., K.C., M.A., and A.I. performed the experiments. K.A. and M.N. analyzed the data. M.N. wrote the MATLAB codes. S.M. and R.O.-S. prepared adeno-associated viruses. K.I. supervised the entire project.

## Competing interests

S.M. owns equity in a company, Gene Therapy Research Institution, that commercializes the use of AAV vectors for gene therapy applications. To the extent that the work in this manuscript increases the value of these commercial holdings, S.M. has a conflict of interest. The other authors declare no competing interests.
