## [Peer Review File · Nature Communications]

REVIEWER COMMENTS

Reviewer #1 (Remarks to the Author):

The authors present results from a novel behavioral paradigm, in which mice enter several distinct contexts and are rewarded according to an implicit hierarchy that was to be inferred by the animals to successfully perform the task. Animals could only infer the implicit hierarchy, when the training of rewarded contexts was randomized towards the end of the training phase and, importantly, when animals slept after the training. By optogenetically targeting the prelimbic cortex (PC), the anterior cingulate cortex (ACC) and the axonal terminals of neurons from the medial entorhinal cortex (MEC) to the ACC, the authors show that the ACC needs to be active during post-training NREM and REM sleep to infer the hierarchy, and that artificially activating the MEC → ACC projections at 4 Hz during REM (but not NREM) sleep makes inference possible, even in the absence of randomization of training trials. These loss-and-gain-of-function experiments are complemented by calcium imaging data of excitatory neurons in the ACC, which were recorded when animals performed the task and during the first two minutes of NREM and REM sleep. The main finding from these data is that neuronal assembly patterns that can be linked to the correct inference in behavior already emerge during the randomization session during training but, reach peak activity strength during post-encoding REM sleep. Co-activation analysis revealed that activation patterns that can be linked to each premise pair, peak during NREM, whereas activity patterns linked to all 4 premise pairs together as well as the inference patterns peak during REM sleep.

The manuscript is clearly written and, overall, the numerous methods used in the experiments appear to be sound, although some details need more clarification (see below). The findings are very compelling in that they delineate different roles of NREM and REM sleep in the formation and shaping of inferential knowledge that, in turn, can be linked to ACC activity during these states. As a side note, I was intrigued by the fact that these findings resemble ideas on which sleep-wake Bayesian program learning algorithms are built upon (I was reminded of the DreamCoder paper by the Tenenbaum-Lab), although the authors do not touch upon this similarity in their manuscript. Some lack of clarity remained, however, regarding the behavioral protocol, which need to be addressed before a final decision is made:

Major points:

1. The authors write “The inferential behavior that appeared in T2 in the inference group was not due to learning during T1 since mice were not provided with sucrose in T1, even when the correct choice was made.” (line 123 - 125). This is an important finding for the understanding of the task. However, some details on the protocol used were not clear:

1. Were mice rewarded for the correct choice when presented with pair B and D in T2, T3? If so, in the following test sessions T3 and T4 mice do not need to use an inference strategy anymore to solve that task.

2. In the methods section (line 513 - 515) it is mentioned that some animals were rewarded during the test session. The animals should not be rewarded at any time during the test, in my understanding? Please clarify in more detail, when animals received sucrose and when not during the test sessions.

3. Could the authors, please, provide also video files showing behavior during test trials? This is a new and highly complex behavioral paradigm, and it would be elucidating to judge the behavior more directly than merely on the basis of summary statistics.

2. In the incomplete hierarchy experiment, the authors present animals with context hierarchies $A > B > C$ and $E > D > C$. I did not understand what a correct inference in this task would be, since the test pair B & D does not form a clear hierarchy. In my view, the absence of a preferred context could also reflect the correct inference that no context is preferable to another.

3. As I understand, the quantification of neural ensembles and reactivation strength are based on the extraction of components from the neuronal Ca^{2+} transients. To relate these abstract measures again to the network level, it would be interesting to know how many neurons per animal contributed to each component (i.e. how many neuronal members does the inference-related pattern have per animal, etc.).

4. There are several details regarding the optogenetic inhibition/activation experiments that are not sufficiently discussed:

1. While the behavioral effects of the optogenetic inhibition in ACC using ArchT under the control of CamKII are compelling, direct readout of the effect in the ACC is missing. For instance, temperature changes in the tissue due to the prolonged light exposure might affect ACC activity more non-specifically as might be concluded from the specificity of the promoter. Also, there are mixed findings on the effect of prolonged activation of ArchT, with one report (Krueger et al, 2020 *Neurobiol Learn Mem.*) showing off-target effects of prolonged inhibition in the hippocampus (using ArchT), leading to a surprising, and unintended, overall increase of hippocampal activity. Based on this, the authors, in my view, cannot conclude that their optogenetic approach indeed only inhibits ACC excitatory activity for the entire sleep duration, without a sufficient readout method (i.e. spike frequency). This shortcoming should at least be discussed in more detail.

2. Why were animals anesthetized to connect them to the optic fibers instead of permanent implantation? Were there any signs that anesthetizing the animals interfered with behavior (such as inertia)?

3. What is the rationale for the 4 Hz (15 ms pulse width) activation protocol? That activating the MEC → ACC projections supports the inference of context hierarchy in the absence of randomization during training, in my view, is the most surprising finding of this work. Were there any specific a priori hypothesis that led to this approach.

Minor points:

1. Figure 1: How is it explained that mice get better (shorter latency-to-choose, higher % of correct trials) over time for test sessions T1 - T4, although they are not reinforced?

2. In Figures 1 c & g, 2d and 5d, mice did not reach the criterion of 80% correct trials in the C & D session. Were they excluded from the analysis?

3. How many trials did the mice perform during the training, reinforcement and randomization sessions? I wonder about the variance in the percentage of correct trials during the last day of training, which only assume 60, 80 or 100%.
4. Figure 1 d and h could be differently organized, if mice perform only 2 test sessions (4 or two trials?) during T1, T2, T3, T4. These figure panels suggest that many more trials are done and leaves the reader wondering why the mice only reach 50% or 100% of correct trials during the inference test.
5. Why do the authors include so many test days and not only T1 and T2? Could T2 affect subsequent test days T3 and T4?

Reviewer #2 (Remarks to the Author):

This is a very exciting paper which addresses a highly topical and important question in a very thorough careful manner, with very convincing findings. The authors trained mice on a transitive inference hierarchy, using locations as the elements in the hierarchy. After training mice were given reinforcement sessions with either the same order as training or a randomised order - they were later tested on inferences - and only those who had the randomisation were above chance. Next, the randomisation paradigm was used with/without sleep deprivation and the sleep deprivation removed the inference learning. Next, optogenetics was used to inhibit activity in ACC, in wake/NREM/REM and when applied in sleep this again removed inference. Next, calcium imaging was used to identify inference related ensembles in the ACC. Next, they looked for neurones that co-activated in multiple pairs of the hierarchy (e.g. A>B, B>C, etc). This showed greater co-activation in all pairs in NREM than REM. When this was re-calculated for activation in all premise and inference pairs it was greater in REM. Finally, the authors used optogenetics to selectively activate ACC neurones receiving entorhinal projections during the task. Activation of these in REM (but not NREM) it boosted inferences. The authors use their findings to propose a model whereby NREM is important for consolidating the learned hierarchy, but REM is needed for facilitating inferences.

I think this is a landmark paper with very exciting findings. I should, however not that I do not work in rodents, and cannot, therefore, offer comment on the exact methods that were used.

There are nevertheless some things which could be improved, and I have listed these below:

- 1) on figure 1, why does sleep deprivation start after 3 days of learning randomization? Surely the sleep that has already occurred would have consolidated this learning, so the sleep deprivation is too late? Also, how is it that just 4 hours of sleep deprivation are enough. I suspect the authors are basing this on prior work - if so please elaborate, because the paradigm choice seems very odd.

2) Why is the randomization condition necessary to promote inference? It is unclear why reinforcement learning would not be enough to allow inference.

3) why did you choose ACC as the main target for excitation of neurones? And why projections from entorhinal to ACC?

4) figure 3c is very confusing - please describe in detail and improve the diagram.

5) Figure 4 a and b are very confusing. You need to explain much more detail in both figure and text. For instance, what do the different colours represent?

6) In general, the task figures (part a on most figures) could be improved - the very small diagram indicating sleep deprivation or injection of a vector is too small and unclear. You really need to make this much bigger and put a text label saying what is happening, and what these tiny diagrams represent..

Reviewer #3 (Remarks to the Author):

Abdou et al present a tour de force of methods investigating inferred knowledge and contributions of sleep. The results are exciting and bring many new insights. Overall, I only have a few comments, mainly on the fact that these results are different than the results in humans and this is not discussed in the paper.

The main comment I have, is that these results are different than the original Ellenbogen et al paper that showed a sleep effect solely in the second order inference not first order as shown here. However, the results in humans have often been questioned and in the field we have heard of failed replications, which was not surprising since the effect of only second and not first ordered inference never really made sense. I found it very interesting how the authors here explain why the second order inference is confounded since A would always be rewarded and E never. I completely agree with them, but in my opinion they should dedicate a new, extended discussion section to this topic (could fit in line 301 onwards). They need to discuss how and why their results differ from Ellenbogen et al (and any other papers using the paradigm) and should also discuss findings about sleep generating insight (wagner et al 2004 etc.) with the number reduction task. There they have recently reported that stage 1 nonrem is sufficient. The mechanisms should be similar so it would be important to discuss.

This sentence is unclear (abstract):

In a transitive inference paradigm, mice gained the inference one day (what does that mean? One day is unclear. You mean not immediately but after delay) , but not shortly, after complete training

In the results, they should explain better what a “choice” is. Entering or 10s? And I don’t understand how you can have 50% correct for one animal data point at test. Its either correct or wrong if there is just one trial for each pair. Or is there more than one trial?

Line 146: Tse et al 2011 and Wang et al 2023 are not lesions but pharmacology. Further, how was the inhibition with opto done? Was there not a max for light on? If the laser is on for the whole sleep bout of one stage, couldn’t it have been too long for NonREM and lead to frying the brain?

For the imaging its not clear in the study design figure what REM1 and REM2 etc are. Can that be adapted in the figure?

304: cherry picking results a bit...REM also often not shown to be important for memory consolidation, and usually it is only emotional and fear memories that show a relationship. (See Danke Hopfer 2009, Genzel 2009, Rasch 2009, also reviewed in Genzel 2015)

We thank the reviewers for the helpful and constructive comments and suggestions. We have responded to all of the reviewers concerns and have modified the manuscript accordingly (all changes are highlighted in red). Also, we have reformatted the manuscript to comply with the style of Nature Communications.

REVIEWER COMMENTS

Reviewer #1 (Remarks to the Author):

The authors present results from a novel behavioral paradigm, in which mice enter several distinct contexts and are rewarded according to an implicit hierarchy that was to be inferred by the animals to successfully perform the task. Animals could only infer the implicit hierarchy, when the training of rewarded contexts was randomized towards the end of the training phase and, importantly, when animals slept after the training. By optogenetically targeting the prelimbic cortex (PC), the anterior cingulate cortex (ACC) and the axonal terminals of neurons from the medial entorhinal cortex (MEC) to the ACC, the authors show that the ACC needs to be active during post-training NREM and REM sleep to infer the hierarchy, and that artificially activating the MEC → ACC projections at 4 Hz during REM (but not NREM) sleep makes inference possible, even in the absence of randomization of training trials. These loss-and-gain-of-function experiments are complemented by calcium imaging data of excitatory neurons in the ACC, which were recorded when animals performed the task and during the first two minutes of NREM and REM sleep. The main finding from these data is that neuronal assembly patterns that can be linked to the correct inference in behavior already emerge during the randomization session during training but, reach peak activity strength during post-encoding REM sleep. Co-activation analysis revealed that activation patterns that can be linked to each premise pair, peak during NREM, whereas activity patterns linked to all 4 premise pairs together as well as the inference patterns peak during REM sleep.

The manuscript is clearly written and, overall, the numerous methods used in the experiments appear to be sound, although some details need more clarification (see below). The findings are very compelling in that they delineate different roles of NREM and REM sleep in the formation and shaping of inferential knowledge that, in turn, can be linked to ACC activity during these states. As a side note, I was intrigued by the fact that these findings resemble ideas on which sleep-wake Bayesian program learning algorithms are built upon (I was reminded of the

DreamCoder paper by the Tenenbaum-Lab), although the authors do not touch upon this similarity in their manuscript. Some lack of clarity remained, however, regarding the behavioral protocol, which need to be addressed before a final decision is made:

We would like to thank the reviewer for highlighting the conceptual similarity between our manuscript and the DreamCoder preprint. We have checked the preprint and found that the authors developed a machine-learning system, called DreamCoder, that relied on a 'wake-sleep' learning algorithm. In this system, problems were learned and replayed, yielding symbolic representations that were used in subsequent new tasks. In our study, temporally-separated experiences were learned during awake state and replayed during sleep, forming hierarchical representations of the learned knowledge which were used subsequently to develop inferential information (Supplementary Figure 10).

Also, we thank the reviewer for the careful reading, positive feedback, and constructive comments, which helped us to increase the quality of the manuscript. We have taken into consideration all of the comments while revising our manuscript as detailed below:

Major points:

- The authors write “The inferential behavior that appeared in T2 in the inference group was not due to learning during T1 since mice were not provided with sucrose in T1, even when the correct choice was made.” (line 123 - 125). This is an important finding for the understanding of the task. However, some details on the protocol used were not clear:
 1. Were mice rewarded for the correct choice when presented with pair B and D in T2, T3? If so, in the following test sessions T3 and T4 mice do not need to use an inference strategy anymore to solve that task.
 - In the 'Inference' group, mice were not rewarded for the correct choice in T2, T3 and T4, therefore the performance observed was due to an inference strategy, not due to direct learning. This information is explained in the initially submitted manuscript (Methods section, line 511-512).
- 2. In the methods section (line 513 - 515) it is mentioned that some animals were rewarded during the test session. The animals should not be rewarded at any time during the test, in my understanding? Please clarify in more detail, when animals received sucrose and when not during the test sessions.
- During the course of our study, we examined the impact of reward on the performance with different approaches:

1- In Figure 1, we would like to prove that the inference observed was not due to direct learning during initial test sessions, so we did not reward mice for the correct choice in the 'Inference' group. We confirmed that rewarding did not affect the performance by providing reward to mice in the 'Reinforcement' & 'Sleep deprivation' groups after correct choice, and mice did not infer correctly in subsequent test sessions.

2- In Figure 2 (optogenetic inhibition), we would like to confirm that the observed failed inference was due to ACC manipulation, not due to lack of motivation from non-rewarded mice after a correct choice. Therefore, we provided mice with reward during testing (keep their motivation) and we found that this reward did not affect the performance in the subsequent test sessions, as indicated with lower percentage of correct choice in NREM & REM groups during T2, T3 & T4.

- Collectively, we found that rewarding or not-rewarding mice did not affect their performance and our conclusion. Thus, reward is not a critical factor in inference emergence.
- In the revised manuscript, we have clarified the above-mentioned points in the 'Methods' section (page 23, lines 532-538) as follows:

“When they made the correct choice, mice were not rewarded in the inference group (Figure 1) and in both awake and sleep groups (Supplementary Fig. 6) to exclude the possibility of direct learning. However, in the remaining experiments, when mice made the correct choice, they were rewarded with a sucrose tablet in context B (Figures 2, 3, 5 & Supplementary Figures 4, 5, 9). In the latter figures (control & manipulation experiments), we provided reward to mice to keep their motivation and to confirm that any failed performance was due to the manipulation, not due to lack of motivation from non-rewarding after correct choice.”

3. Could the authors, please, provide also video files showing behavior during test trials? This is a new and highly complex behavioral paradigm, and it would be elucidating to judge the behavior more directly than merely on the basis of summary statistics.

- Based on the reviewer's comment, we have included supplementary video files showing the training trial in B & C premise pair (Supplementary Video 1) and the inferential behavior during T2 (Supplementary Video 2) in our revised manuscript.
- In the incomplete hierarchy experiment, the authors present animals with context hierarchies $A > B > C$ and $E > D > C$. I did not understand what a correct inference in this task would be, since the test pair B & D does not form a clear

hierarchy. In my view, the absence of a preferred context could also reflect the correct inference that no context is preferable to another.

- The reviewer's understanding is correct. In the incomplete hierarchy experiment, mice did not prefer any context during B & D test, which is considered correct performance.
 - Based on the reviewer's comment, we have modified the y-axis in Supplementary Figure 5d-g in the revised manuscript. Accordingly, we have modified the corresponding explanation in the text (page 7, line 141-142).
- As I understand, the quantification of neural ensembles and reactivation strength are based on the extraction of components from the neuronal Ca²⁺ transients. To relate these abstract measures again to the network level, it would be interesting to know how many neurons per animal contributed to each component (i.e. how many neuronal members does the inference-related pattern have per animal, etc.).
- In the revised manuscript, we have followed the reviewer's suggestion and have incorporated the ensemble size (number of neurons contributed in each pattern) as new panel (Figure 3j).

- There are several details regarding the optogenetic inhibition/activation experiments that are not sufficiently discussed:
1. While the behavioral effects of the optogenetic inhibition in ACC using ArchT under the control of CamKII are compelling, direct readout of the effect in the ACC is missing. For instance, temperature changes in the tissue due to the prolonged light exposure might affect ACC activity more non-specifically as might be concluded from the specificity of the promoter. Also, there are mixed findings on the effect of prolonged activation of ArchT, with one report (Krueger et al,

2020 *Neurobiol Learn Mem.*) showing off-target effects of prolonged inhibition in the hippocampus (using ArchT), leading to a surprising, and unintended, overall increase of hippocampal activity. Based on this, the authors, in my view, cannot conclude that their optogenetic approach indeed only inhibits ACC excitatory activity for the entire sleep duration, without a sufficient readout method (i.e. spike frequency). This shortcoming should at least be discussed in more detail.

- Several previous reports have used prolonged activation protocol to ArchT in hippocampus and cortical regions and they showed diminished activity and decreased in the neuronal firing rate¹⁻².

References

1. Miyamoto, D., et al. Top-down cortical input during NREM sleep consolidates perceptual memory. **Science** 352, 1315-1318 (2016)
 2. Wally, M., et al. A short-term memory trace persists for days in the mouse hippocampus. **Communications Biology** 5, 1168 (2022)
- Furthermore, we believe that our optogenetic inhibition protocol did not cause brain damage or off-target effects due to several reasons:
 - 1- In Supplementary Figure 6, prolonged light illumination on PL during awake and sleep states did not affect both the learned & inferred information. This result excludes the possibility of off-target effects (widespread inhibition) since ACC is anatomically adjacent to PL and if the light illumination reached to ACC, it would have disrupted the inferential behavior as observed in Figure 2.
 - 2- After prolonged light exposure during NREM sleep, mice could perform well during testing the premise pairs on day 28, 29 (Supplementary Figure 2c). Mice performance during testing the premise pairs after long NREM inhibition was comparable to that observed during training stage (before the inhibition).
 - 2- In Figure 2, the same inhibition protocol was applied during awake state and both the learned and inferred knowledge were not affected (Figure 2 & Supplementary Figure 2c). These results suggest that the inhibition protocol did not cause damage to ACC.
 - In the revised manuscript, we have followed the reviewer's suggestion and have added the following paragraph in the discussion section (page 14, lines 315-320):

Recent study has reported that prolonged light activation to ArchT on hippocampus may lead to rebound increase of hippocampal activity⁴⁶. Therefore, the observed failed inference in Figure 2 might be due to either decrease in ACC activity or rebound overall ACC activation which might interfere with ACC computations. The

latter possibility is less likely to be the reason for failed inference since ACC activation during REM sleep boosted inference emergence (Figure 5e).

2. Why were animals anesthetized to connect them to the optic fibers instead of permanent implantation? Were there any signs that anesthetizing the animals interfered with behavior (such as inertia)?

- Technically, we don't have the set-up for permanent fiber implantation and the optic fiber is fragile, so it might be broken if attached to mouse head for several days. Therefore, we attached and detached the optic fiber before and after the manipulation, respectively.
- There was no effect for the anesthesia on the mice behavior because in the optogenetic inhibition & activation experiments, the anesthesia was performed after the behavior session (not before) and the next behavior session was performed at least after 1 day, when we were sure that the anesthesia effect was gone. Also, our study includes several control groups that showed successful inferential behavior regardless being anesthetized; for example,
 - 1- Awake group in Figure 2
 - 2- Awake, NREM, REM groups in Supplementary Figure 6
 - 3- Mice underwent Ca²⁺ imaging experiment in Figure 3
 - 4- REM group in Figure 5

3. What is the rationale for the 4 Hz (15 ms pulse width) activation protocol? That activating the MEC → ACC projections supports the inference of context hierarchy in the absence of randomization during training, in my view, is the most surprising finding of this work. Were there any specific a priori hypothesis that led to this approach.

- In the initially submitted manuscript (lines 307-311), we discussed the rationale for using 4 Hz stimulation protocol. Briefly, it has been demonstrated that
 - 1- The 4 Hz oscillations are dominant in PFC
 - 2- The 4 Hz oscillations were phase coupled with theta rhythm (4-8 Hz) which is dominant in REM sleep, supporting flexible information processing (Fujisawa, S. & Buzsaki, G, Neuron, 2011)
- Furthermore, previous study showed that 4 Hz optogenetic stimulation protocol to PFC could induce the recall of previously acquired memory (Kitamura et al., Science, 2017)

References

Fujisawa, S. & Buzsaki, G. A 4 Hz oscillation adaptively synchronizes prefrontal, VTA, and hippocampal activities. Neuron 72, 153-165 (2011).

<https://doi.org/10.1016/j.neuron.2011.08.018>

Kitamura, T. et al. Engrams and circuits crucial for systems consolidation of a memory. Science 365, 73-78 (2017).

Minor points:

1. Figure 1: How is it explained that mice get better (shorter latency-to-choose, higher % of correct trials) over time for test sessions T1 - T4, although they are not reinforced?

- In 'Reinforcement' group, the improvement in performance was not statistically significant and did not reach to the performance of 'Randomization' & 'Inference' groups, so this improvement was not conclusive. However, we believe that we observed this gradual improvement because some mice in the 'Reinforcement' group learnt during consecutive test sessions since they were rewarded in T1, T2 and T3. However, even rewarded, they did not make significant inference compared to the non-rewarded 'Inference' and 'Randomization' group.

2. In Figures 1 c & g, 2d and 5d, mice did not reach the criterion of 80% correct trials in the C & D session. Were they excluded from the analysis?

- These mice were not excluded from the analysis to avoid any bias in the results.
- Although some mice had 60% correct trials in C & D session, they made successful inference in the test sessions. This finding might suggest that criterion of 80% correct trials is quite high.

3. How many trials did the mice perform during the training, reinforcement and randomization sessions? I wonder about the variance in the percentage of correct trials during the last day of training, which only assume 60, 80 or 100%.

- 5 trials for each premise pair during the training, reinforcement and randomization sessions. This information was mentioned in the 'Methods' section in the initially submitted manuscript.
- In the revised manuscript, we have added the number of trials in Figure 1 and its legend to make it clear for the readers that every premise pairs session includes 5 trials for each pair & a test session includes 2 trials for each test. Therefore, percentage of correct trials during premise pair session (5 trials) should be 0%, 20%, 40%, 60%, 80%, 100%, while during test session (2 trials) should be 0%, 50%, 100%.

4. Figure 1 d and h could be differently organized, if mice perform only 2 test sessions (4 or two trials?) during T1, T2, T3, T4. These figure panels suggest that many more trials are done and leaves the reader wondering why the mice only reach 50% or 100% of correct trials during the inference test.

- In the revised manuscript, we have followed the reviewer's suggestion and have added the number of trials in Figure 1 and its legend to make it clear for the readers that every premise pairs session includes 5 trials for each pair & a test session includes 2 trials for each test. Therefore, percentage of correct trials during premise pair session (5 trials) should be 0%, 20%, 40%, 60%, 80%, 100%, while that percentage during test session (2 trials) should be 0%, 50%, 100%.

5. Why do the authors include so many test days and not only T1 and T2? Could T2 affect subsequent test days T3 and T4?

- We believe that T1 and T2 were enough to show the emergence of sleep-dependent inference and its temporal mechanism. However, we included T3 and T4 for several reasons:
 - 1- To increase the total number of inference trials since each test session includes 2 trials.
 - 2- To demonstrate the consistency of the inferential behavior across different days, that's why we did not increase the number of trials in one session per day.
 - 3- To test how long the inference building ability would last. We found, in the absence of reward in test session, mice could infer correctly till T4 (3 days after the last day of randomized training).
- We believe that T2 did not affect performance in T3 and T4 because mice did not receive reward (feedback) after their choice in T1 and T2 (please refer to the first comment, point #2).

Reviewer #2 (Remarks to the Author):

This is a very exciting paper which addresses a highly topical and important question in a very thorough careful manner, with very convincing findings. The authors trained mice on a transitive inference hierarchy, using locations as the elements in the hierarchy. After training mice were given reinforcement sessions with either the same order as training or a randomised order - they were later

tested on inferences - and only those who had the randomisation were above chance. Next, the randomisation paradigm was used with/without sleep deprivation and the sleep deprivation removed the inference learning. Next, optogenetics was used to inhibit activity in ACC, in wake/NREM/REM and when applied in sleep this again removed inference. Next, calcium imaging was used to identify inference related ensembles in the ACC. Next, they looked for neurons that co-activated in multiple pairs of the hierarchy (e.g. A>B, B>C, etc). This showed greater co-activation in all pairs in NREM than REM. When this was re-calculated for activation in all premise and inference pairs it was greater in REM. Finally, the authors used optogenetics to selectively activate ACC neurons receiving entorhinal projections during the task. Activation of these in REM (but not NREM) it boosted inferences. The authors use their findings to propose a model whereby NREM is important for consolidating the learned hierarchy, but REM is needed for facilitating inferences.

I think this is a landmark paper with very exciting findings. I should, however not that I do not work in rodents, and cannot, therefore, offer comment on the exact methods that were used.

There are nevertheless some things which could be improved, and I have listed these below:

We thank the reviewer for the careful reading, positive feedback, and constructive comments, which helped us to increase the quality of the manuscript. We have taken into consideration all of the comments while revising our manuscript as detailed below:

1) on figure 1, why does sleep deprivation start after 3 days of learning randomization? Surely the sleep that has already occurred would have consolidated this learning, so the sleep deprivation is too late? Also, how is it that just 4 hours of sleep deprivation are enough. I suspect the authors are basing this on prior work - if so please elaborate, because the paradigm choice seems very odd.

- We have done sleep deprivation (SD) on the last day of randomization stage for several reasons:
 - 1- We preferred to avoid accumulated effect from successive SD sessions. If we started the SD from the first day of randomization, we would have 4 SD sessions before T2 (inference test).
 - 2- We believe that the sleep after complete randomization is necessary for building the hierarchy, hence extracting the inferential information. Complete

randomization requires the interaction between all premise pairs together which happens on the last day of randomization since 1 cycle of complete randomization takes 4 days.

3- During the establishment of the ‘transitive inference’ behavior task, we conducted a preliminary experiment (not shown in our manuscript) where mice were exposed to only 2 days of randomized training (incomplete) instead of 4 days. Mice did not infer correctly during T2 (inference test) session, therefore we decided to increase the number of days to perform complete interaction between all premise pairs together.

In the revised manuscript, we have added the above figure as (Supplementary Figure 3) and added the corresponding explanation in text (page 5-6, lines 104-110).

“To examine the temporal contribution of randomization stage in inference emergence, mice were trained on the same behavior protocol, but with only 2-days randomization (incomplete randomization) (Supplementary Fig. 3a). Mice successfully learned the premise pairs (Supplementary Fig. 3b); however, they did not make correct inference during testing sessions (Supplementary Fig. 3b). This result suggests that ensuring complete interaction between all premise pairs is necessary for inference emergence. Complete interaction between the learned pairs occurred after 4 days of randomized training (on day 24).”

- Regarding the duration of sleep deprivation protocol (4 hours): Before starting this experiment, we reviewed the literature studying the effect of sleep deprivation on rodent behavior and we found that sleep deprivation for 2-6 hours impaired some of the brain functions¹⁻⁴. Based on this, we utilized the 4hr sleep deprivation protocol in our study.

References

- 1- Sawangjit, A., et al. The hippocampus is crucial for forming non-hippocampal long-term memory during sleep. *Nature* 109, 564 (2018)
- 2- Vecsey, C., et al. Sleep deprivation impairs cAMP signaling in the hippocampus. *Nature* 461 (2009)
- 3- Bolsius, Y., et al. Recovering object-location memories after sleep deprivation-induced amnesia. *Current Biology* 33, 298-308 (2023)
- 4- Prince, T., et al. Sleep deprivation during a specific 3-hour time window post-training impairs hippocampal synaptic plasticity and memory. *Neurobiology of Learning and Memory* 109, 122-130 (2014)

2) Why is the randomization condition necessary to promote inference? It is unclear why reinforcement learning would not be enough to allow inference.

- It was an interesting finding for us. We believe that inference emerged after organizing the learned information in complete hierarchy (A>B>C>D>E) which means that all premise pairs should connect and interact together.
- In reinforcement learning, premise pairs were not interconnected since (A>B & B>C) pairs were introduced in different days than (C>D & D>E) pairs across the behavior timeline, which might hinder the hierarchy formation, hence inference emergence.
- In randomized learning, premise pairs were interconnected together since every premise pair was co-presented with the other premise pairs, ensuring complete interaction between all of them. Therefore, this might facilitate organizing the premise pairs in hierarchy, hence building the inferential knowledge.

3) why did you choose ACC as the main target for excitation of neurons? And why projections from entorhinal to ACC?

- Choosing ACC & MEC as candidate regions for optogenetic activation was done based on both previous studies and the results obtained during the course of our study.
 - a- As mentioned in the initially submitted manuscript, several previous studies have proposed that medial prefrontal cortex (mPFC) might be involved in computing inferential reasoning (Reference no. 4, 5, 36, 37, 38 in the revised manuscript)
 - b- We investigated 2 mPFC sub-regions (PL & ACC) and we found that ACC, but not PL, activity was necessary for the inference emergence (Supplementary Fig. 5). Then, we identified how ACC neurons compute inferential knowledge by Ca²⁺ imaging. Thus, we targeted ACC neurons in the gain-of-function experiment to complete the whole picture of the story.

c- MEC has been proposed to be involved in the processing of inferential information (Reference no. 42, 43) and has a direct synaptic connection with ACC (*Kitamura, T. et al, Science, 2017*).

- Collectively, we thought that ACC coactivity code (observed in Figure 4) could rely on synaptic inputs from upstream region (MEC) which is thought to be involved in the transitive inference.
- 4) Figure 3c is very confusing - please describe in detail and improve the diagram.
- As requested, in the revised manuscript, we have modified Figure 3c and added some details in the figure legend.
- 5) Figure 4 a and b are very confusing. You need to explain much more detail in both figure and text. For instance, what do the different colors represent?
- In the revised manuscript, we have followed the reviewer's comment and have modified (Figure 4a, b) to be more clear. Also, we have added explanation to the colors in the figure legend.
 - Different colors represent different neurons constituting different patterns which represent different sessions. Simply, each color denotes group of neurons representing specific behavioral session. In the revised manuscript, this information has been adapted in the figure.
- 6) In general, the task figures (part a on most figures) could be improved - the very small diagram indicating sleep deprivation or injection of a vector is too small and unclear. You really need to make this much bigger and put a text label saying what is happening, and what these tiny diagrams represent.
- In the revised manuscript, we have followed the reviewer's comment and have enlarged some panels in the following figures (Figures 2, 3, 4, 5 & Supplementary Figure 6).

Reviewer #3 (Remarks to the Author):

Abdou et al present a tour de force of methods investigating inferred knowledge and contributions of sleep. The results are exciting and bring many new insights. Overall, I only have a few comments, mainly on the fact that these results are different than the results in humans and this is not discussed in the paper. We thank the reviewer for the careful reading, positive feedback, and constructive

comments, which helped us to increase the quality of the manuscript. We have taken into consideration all of the comments while revising our manuscript as detailed below:

The main comment I have, is that these results are different than the original Ellenbogen et al paper that showed a sleep effect solely in the second order inference not first order as shown here. However, the results in humans have often been questioned and in the field, we have heard of failed replications, which was not surprising since the effect of only second and not first ordered inference never really made sense. I found it very interesting how the authors here explain why the second order inference is confounded since A would always be rewarded and E never. I completely agree with them, but in my opinion, they should dedicate a new, extended discussion section to this topic (could fit in line 301 onwards). They need to discuss how and why their results differ from Ellenbogen et al (and any other papers using the paradigm) and should also discuss findings about sleep generating insight (wagner et al 2004 etc.) with the number reduction task. There they have recently reported that stage 1 nonrem is sufficient. The mechanisms should be similar so it would be important to discuss.

- We have followed the reviewer's suggestion and have discussed the above-mentioned point in our revised manuscript. We have added the following paragraph in (page 13-14, lines 301-314):
"Our demonstration is consistent with previous reports showing the crucial role of sleep in inspiring insight^{2,45} and extracting inferential information³. A recent study on human subjects has shown that stage 1 NREM sleep might be sufficient to inspire creative problem solving⁴⁵, while our study showed that NREM sleep was necessary, but not sufficient to build inferential knowledge. This discrepancy might be attributed to differences between both tasks, in terms of task modalities and task timeline. Also, there are differences between mouse and human brain that put predictable limitations on relating data across species. These notions may explain the differences between the inference observed in our study and that observed in a previous study on human subjects³. Both studies have showed that inference was not developed shortly after learning the premise pairs. However, upon examining the first-order & second-order inference, they found that only second-order inference was sleep-dependent while first-order inference was sleep-independent³. Conversely, we proved that first-order inference was sleep-dependent using cellular recording and causal manipulation experiments."

This sentence is unclear (abstract):

In a transitive inference paradigm, mice gained the inference one day (what does that mean? One day is unclear. You mean not immediately but after delay), but not shortly, after complete training

- In this sentence, we would like to highlight when the inferential behavior was observed. The sentence means ‘mice gained the inference one day after complete training’.
- In the revised manuscript, we have modified the sentence in the abstract to be clearer “In a transitive inference paradigm, mice gained the inference 1 day after, but not shortly after, complete training”.

In the results, they should explain better what a “choice” is. Entering or 10s? And I don’t understand how you can have 50% correct for one animal data point at test. Its either correct or wrong if there is just one trial for each pair. Or is there more than one trial?

- In our study, the ‘Choice’ includes 2 steps: (1) entering to the context and (2) waiting 10 seconds, then mice were rewarded in the correct context.
- In the revised manuscript, we have added the following sentence in (page 4, line 81- 82) “The correct choice requires 2 steps: (1) entering to the assigned correct context and (2) staying in the assigned context for a consecutive 10 sec, then mice received a sucrose tablet”.
- Every test session consists of 2 trials, so the overall performance should be 0% or 50% or 100% and we have stated this information in the ‘Methods’ section in the initial manuscript. Also, in the revised manuscript, we have added the number of trials in Figure 1d and its legend.

Line 146: Tse et al 2011 and Wang et al 2023 are not lesions but pharmacology. Further, how was the inhibition with opto done? Was there not a max for light on? If the laser is on for the whole sleep bout of one stage, couldn’t it have been too long for NonREM and lead to frying the brain?

- In the initially submitted manuscript (Methods section), we have stated that optogenetic inhibition was done by the delivery of continuous light during specific sleep stage bout. The maximum duration of light ON during each bout was 15 minutes, with maximum total duration of 1.5 hour / mouse.
- We believe that the inhibition protocol did not cause brain frying or decline in brain function due to several reasons:
 - 1- After applying this protocol during NREM sleep, mice could perform well during testing the premise pairs on day 28, 29 (Supplementary Figure 2c). Mice

performance during testing the premise pairs after long NREM inhibition was comparable to that observed during training stage (before the inhibition).

2- The same inhibition protocol was applied during awake state and both the learned and inferred knowledge were not affected (Figure 2 & Supplementary Figure 2c). These results suggest that the inhibition protocol did not cause damage to ACC.

- Furthermore, several previous reports have used similar stimulation protocol with duration of light ON ranging from 1 hour to 5 hours without reporting any brain damage¹⁻⁴.

References

1- Wally, M., et al. A short-term memory trace persists for days in the mouse hippocampus. *Communications Biology* 5, 1168 (2022)

2- Konadhode, R., et al. Optogenetic stimulation of MCH neurons increases sleep. *Journal of Neuroscience* 33(25):10257–10263 (2013)

3- Rolls, A., et al. Optogenetic disruption of sleep continuity impairs memory consolidation. *Proc Natl Acad Sci USA* 32, 13305-13310 (2011)

4- Carter, M., et al. Tuning arousal with optogenetic modulation of locus coeruleus neurons. *Nature Neuroscience* 13, 12 (2010)

For the imaging it is not clear in the study design figure what REM1 and REM2 etc are. Can that be adapted in the figure?

- In the revised manuscript, we have followed the reviewer's suggestion and modified Figure 3c to adapt NREM 1, REM 1, awake, NREM 2 & REM 2 imaging sessions.
- In the initially submitted manuscript & the revised one (in Methods section & in the legend of Figure 3), we have stated that NREM 1 & REM 1 represent the sleep sessions after arena habituation, while NREM 2 & REM 2 represent the sleep sessions after Test 1 session.

304: cherry picking results a bit...REM also often not shown to be important for memory consolidation, and usually it is only emotional and fear memories that show a relationship. (See Danke Hopfer 2009, Genzel 2009, Rasch 2009, also reviewed in Genzel 2015)

- A recent study, published in Science 2016, showed that REM sleep is important for consolidating both emotional and spatial (contextual) memories. This reference is cited in the initially submitted manuscript & the revised one (reference no. 8).

- In the revised manuscript (page 14, line 322-324), we have revised the sentence to be “Prior investigations into the role of REM sleep in cognition has found that it is necessary for consolidating both contextual and emotional memories which involve amygdala-related networks”. Also, we have added the suggested reference (Genzel 2015) as in-text citation and to the reference list (reference no. 48).

REVIEWER COMMENTS

Reviewer #1 (Remarks to the Author):

In the revised manuscript the authors have clarified much of the ambiguities regarding the behavioral protocol. Indeed, I am very positive with this revision. Some issues, however, remain:

1. I still do not fully understand the necessity of rewarding the correct choice (entering and spending time in context B) during the test phase. The authors argue that this was done to keep the animals motivated to perform the task, although it was shown in the initial Inference Group that the animals performed the task well without any additional reward. As a result, the additional reward during test trials makes the conclusion from the other experimental groups more complex: The inference that B>D learned during the randomized learning trials is behaviorally stronger than the behavior resulting from the reward of B over D in the test trials. This is an interesting finding and should be explicitly discussed in the Discussion section to make it easier to understand the experiments.
2. Thank you also for providing one example video from the test phase. I was wondering why the arena was differently positioned in the room during the test phase compared to the video of the learning trial? Is this because two different animals are shown? To allow for a thorough judgment of this novel and rather complex behavioral task, the authors should not just present one example video but, optimally, make accessible the entire behavioral data set.
3. I don't think that the conclusion "These results demonstrate that extracting inference requires the organization of previously acquired knowledge into a hierarchical order" from the incomplete hierarchy experiment is warranted. As the authors confirmed, the correct inference in this task would be no preference for one context. Animals that do not show a preference for one context could therefore either have learned the correct inference (that no context is preferable) or could have failed to learn this inference. In both cases there would be no context preference. As a result any conclusion about what animals have actually learned about the hierarchies presented during the learning trials in this experiment is impossible. It rather shows that to test inference, the behavioral readout of successful inference learning needs to differ from the behavioral readout of failed inference learning.

Reviewer #2 (Remarks to the Author):

I thank you for your revision and explanations. You have answered almost all my points, but I do think your figures still need work. Please realise that I am asking this because I would like your readers to be able to understand what you have done, and without improving these (especially figure 4) I don't think many will.

4) Figure 3c is very confusing - please describe in detail and improve the diagram.

- As requested, in the revised manuscript, we have modified Figure 3c and added some details in the figure legend.

REVIEWER: Thank you. But what do the colours represent here? Eg. $A > B$ and $B > C$ are in different colours, etc. Why do you have of these pairs for 11-12 and then only one each for 13 and 14? If this figure isn't clear people won't be able to understand your method.

5) Figure 4 a and b are very confusing. You need to explain much more detail in both figure and text. For instance, what do the different colours represent?

- In the revised manuscript, we have followed the reviewer's comment and have modified (Figure 4a, b) to be more clear. Also, we have added explanation to the colours in the figure legend.
- Different colours represent different neurons constituting different patterns which represent different sessions. Simply, each colour denotes group of neurons representing specific behavioural session. In the revised manuscript, this information has been adapted in the figure.

REVIEWER: Thank you, but this is still not very clear. For instance, in figure 4a, it is not clear how the values for pairwise co-activity relate to the figure above (e.g. in surely this shows no shared activity (no overlap in which neurons are active for different pairs) until box 6, where I see one overlap) if there is more co-activity how is it represented here? In Fig4b, you say the red is for high co-activity, while blue is for low. Patterns are on the x and y axes. So does this mean that the same neurons involved in pattern 1 are involved in pattern 3 at a high level (what is the cut-off for high/low)? Same for patterns 4 and 6? What are the patterns exactly? I really think you need this info in the figure legend in order for readers to understand.

Reviewer #3 (Remarks to the Author):

I still believe this is an important article however I am disappointed how little the authors changed the article in response to comments. If something is unclear to the reviewers, it should be added to the manuscript.

There are especially two comments not addressed well.

1, It may be counter to the nature of the authors to directly point out issues in the literature, but it is very important to do so for us to advance as a field. Thus it is critical that the authors rediscuss when

mentioning the human study why they chose not to look at the second order interference. This section should be more critical since some of us in the field are very aware that the Ellenbogen paper has not been replicated by others. In general a more in depth discussion should be added and not just writing that rodent and human brains are different...why then run an analogue experiment.

2, it should be added as caveat that 15min light on is very long and could have led to damage.

REVIEWER COMMENTS

We thank the reviewers for the helpful and constructive comments and suggestions. We have responded to all of the reviewers concerns and have modified the manuscript accordingly with red color. Also, we have reformatted the manuscript to comply with the style of Nature Communications.

Reviewer #1 (Remarks to the Author):

In the revised manuscript the authors have clarified much of the ambiguities regarding the behavioral protocol. Indeed, I am very positive with this revision.

We would like to thank the reviewer for positive feedback. We have taken into consideration all of the comments while revising our manuscript as detailed below:

Some issues, however, remain:

1. I still do not fully understand the necessity of rewarding the correct choice (entering and spending time in context B) during the test phase. The authors argue that this was done to keep the animals motivated to perform the task, although it was shown in the initial Inference Group that the animals performed the task well without any additional reward. As a result, the additional reward during test trials makes the conclusion from the other experimental groups more complex: The inference that $B > D$ learned during the randomized learning trials is behaviorally stronger than the behavior resulting from the reward of B over D in the test trials. This is an interesting finding and should be explicitly discussed in the Discussion section to make it easier to understand the experiments.

- We believe that this is a point of strength in our study and does not complicate our conclusion, instead, supports it.
 - 1- In control and manipulation experiments, the observed failed inference even in the presence of reward strengthens our conclusion, since providing reward during test sessions did not overcome the effect of our manipulation (Figure 2e,f).
 - 2- In these experiments, the behavioral outcome (especially T1) was failed-inference, which means that mice did not get reward enough to learn $B > D$ since each test session consists of only 2 trials. Therefore, percentage of correct trials (rewarded) in case of failed inference is 0% or 50%, which means mice receive only 1 reward pellet per day. In contrast, during training phase (premise pairs), mice were exposed to 5 trials per day for 4 days to learn preferring one context over the other.
- Based on the reviewer's suggestion, we have discussed the above point in the discussion section in our revised manuscript by adding the following paragraph

(page 14-15, lines 333-347): “In Figure 1, when mice made the correct choice, they were not rewarded to exclude the possibility of direct learning of $B > D$ during test sessions. However, in the optogenetic inhibition experiment (Figure 2), mice were rewarded in context B during test sessions to confirm that the observed failed-inference was due to manipulating the offline ACC computations, not due to lack of motivation from non-rewarding after a correct choice. Providing reward during test sessions did not overcome the effect of manipulation, which strengthen our conclusion. Since test sessions consist of only 2 trials and percentage of correct trials (rewarded) in case of failed inference is 0% or 50%, therefore mice would receive only 1 reward pellet per day that is not sufficient for mice to prefer context B over D. The successful inference in the absence of reward (Figure 1) along with failed inference in the presence of reward (Figure 2) during test session indicates that reward during test sessions is not necessary for the inferential behavior. These notions can further explain the results that the inferential behavior ($B > D$) computed from arranging the randomized pairs in hierarchy (Figure 1d,e) was more evident than the behavior resulting from direct rewarding context B over D (Figure 2 & Supplementary Figure 5).”

2. Thank you also for providing one example video from the test phase. I was wondering why the arena was differently positioned in the room during the test phase compared to the video of the learning trial? Is this because two different animals are shown? To allow for a thorough judgment of this novel and rather complex behavioral task, the authors should not just present one example video but, optimally, make accessible the entire behavioral data set.

- In all experiments, the arena was differently positioned in the room across sessions to avoid the effect of distal cues on performance. We would like to be sure that the observed performance was only due to the context preference, not the preference of any other spatial cue outside the arena.
- We have added the above-mentioned point in the revised manuscript (Page 21, lines 508-510).
- Our study includes more than 3000 behavioral videos, so it takes very long time to upload all videos. Therefore, we uploaded representative videos from training trials and test trials with the 2 different arena orientations. We believe that the reviewer and readers could imagine the behavioral task along with detailed explanation in the Methods section.

3. I don't think that the conclusion “These results demonstrate that extracting inference requires the organization of previously acquired knowledge into a

hierarchical order” from the incomplete hierarchy experiment is warranted. As the authors confirmed, the correct inference in this task would be no preference for one context. Animals that do not show a preference for one context could therefore either have learned the correct inference (that no context is preferable) or could have failed to learn this inference. In both cases there would be no context preference. As a result, any conclusion about what animals have actually learned about the hierarchies presented during the learning trials in this experiment is impossible. It rather shows that to test inference, the behavioral readout of successful inference learning needs to differ from the behavioral readout of failed inference learning.

- We agree with the reviewer that the behavioral readout of correct inference differs according to the hierarchy status. Therefore, we have followed the reviewer’s suggestion and removed the following sentence from the revised manuscript: “These results demonstrate that extracting inference requires the organization of previously acquired knowledge into a hierarchical order.”

Reviewer #2 (Remarks to the Author):

I thank you for your revision and explanations. You have answered almost all my points, but I do think your figures still need work. Please realise that I am asking this because I would like your readers to be able to understand what you have done, and without improving these (especially figure 4) I don't think many will.

We thank the reviewer for the positive feedback. We have taken into consideration all of the comments while revising our manuscript as detailed below:

4) Figure 3c is very confusing - please describe in detail and improve the diagram.

- As requested, in the revised manuscript, we have modified Figure 3c and added some details in the figure legend.

REVIEWER: Thank you. But what do the colours represent here? Eg. $A > B$ and $B > C$ are in different colours, etc. Why do you have of these pairs for 11-12 and then only one each for 13 and 14? If this figure isn't clear people won't be able to understand your method.

- In Figure 3c, to avoid confusion and misinterpretation, each session is represented by a distinct color. This color-coding scheme allows readers to easily track the performance of each session throughout the data. We used this color coding across all figures (from Figure 1) for consistency. We have added the

following sentence in the revised manuscript (in the legend of Figure 1; lines 954-955): “premise pairs were consistently color-coded across all figures to facilitate tracking each pair without confusion.”

- In Figure 3c, the training protocol in (days 11 & 12) is similar to that employed in Figure 1 & 2. However, in days 13 & 14, the 2 pairs were separated to extract specific representations for a particular premise pair without interfering with the other pair.
- We have followed the reviewer’s suggestion and have added the following sentence in the revised manuscript (in the legend of Figure 3c; lines 1001-1004): “During training stage, Ca^{2+} transients were collected for each premise pair in a separate day (days 13, 14, 17, 18) to extract specific representations for a particular premise pair without interfering with the other pair.”

5) Figure 4 a and b are very confusing. You need to explain much more detail in both figure and text. For instance, what do the different colours represent?

- In the revised manuscript, we have followed the reviewer’s comment and have modified (Figure 4a, b) to be clearer. Also, we have added explanation to the colours in the figure legend.
- Different colours represent different neurons constituting different patterns which represent different sessions. Simply, each colour denotes group of neurons representing specific behavioural session. In the revised manuscript, this information has been adapted in the figure.

REVIEWER: Thank you, but this is still not very clear. For instance, in figure 4a, it is not clear how the values for pairwise co-activity relate to the figure above (e.g. in surely this shows no shared activity (no overlap in which neurons are active for different pairs) until box 6, where I see one overlap) if there is more co-activity how is it represented here? In Fig4b, you say the red is for high co-activity, while blue is for low. Patterns are on the x and y axes. So does this mean that the same neurons involved in pattern 1 are involved in pattern 3 at a high level (what is the cut-off for high/low)? Same for patterns 4 and 6? What are the patterns exactly? I really think you need this info in the figure legend in order for readers to understand.

- Based on the reviewer’s suggestion, we have modified Figure 4 in the revised manuscript and the figure legend accordingly.
- We have added a new figure panel (Fig. 4a) that provides a detailed explanation of the coactivity calculation method. This makes the previous panel (a) become panel (b). Additionally, an equation deriving coactivity index from the calculated

coactivity values was included in the Fig. 4b. The legend for Figure 4 has been revised and expanded.

- The previous version of Fig. 4b depicted coactivity values higher than those calculated based on shuffled data. However, because this point is indicated in Fig 4d,e and also because it is difficult to understand for readers as pointed out by the reviewers, we have removed previous Fig.4b.

Reviewer #3 (Remarks to the Author):

I still believe this is an important article however I am disappointed how little the authors changed the article in response to comments. If something is unclear to the reviewers, it should be added to the manuscript.

We thank the reviewer for believing in the importance of our manuscript. We have taken into consideration all of the comments while revising our manuscript as detailed below:

There are especially two comments not addressed well.

1, It may be counter to the nature of the authors to directly point out issues in the literature, but it is very important to do so for us to advance as a field. Thus, it is critical that the authors re-discuss when mentioning the human study why they chose not to look at the second order interference. This section should be more critical since some of us in the field are very aware that the Ellenbogen paper has not been replicated by others. In general, a more in-depth discussion should be added and not just writing that rodent and human brains are different...why then run an analogue experiment.

- Since we have 5 contexts in the hierarchy, we could test only first-order inference (B & D), but any other test would be either non-inference test or testing the learned premise pairs.
- Based on the reviewer's suggestion, we have discussed this point and we believe that the following discussion part (Page 14, lines 311-320) covers how and why our results differ from Ellenbogen paper, in addition to why we did not focus on second-order inference:

“Both studies have showed that inference was not developed shortly after learning the premise pairs. However, upon examining the first-order & second-order inference, they found that only second-order inference was sleep-dependent while first-order inference was sleep-independent³. Conversely, we demonstrated that first-order inference (B & D) was sleep-dependent using cellular recording and causal manipulation experiments. This finding may suggest that more higher-order inference is even more likely to be sleep-dependent; therefore, we opted

not to test the sleep dependence of second-order inference. On the other hand, non-inference relational memory (A & E) was independent of sleep, likely because A was always rewarded and E was never rewarded.”

2, it should be added as caveat that 15min light on is very long and could have led to damage.

- We have followed the reviewer’s suggestion and clarified this point in our revised manuscript. We have added the following paragraph to the discussion section (Pages 14-15, lines 326-332):

“On the other hand, previous reports showed that prolonged light stimulation may cause heat-induced neuronal damage which alters cell behavior and neuronal excitability^{47,48}. However, we confirmed behaviorally that the optogenetic protocol used in our study did not cause brain damage since both the learned and inferred knowledge were not affected by prolonged light stimulation to ACC during awake state. Also, mouse performance during testing the premise pairs after long NREM inhibition was comparable to that observed during the training stage (Figure 2d & Supplementary Figure 2c).”

References:

47. Peixoto, H. M., Cruz, R. M. S., Moulin, T. C. & Leão, R. N. Modeling the Effect of Temperature on Membrane Response of Light Stimulation in Optogenetically-Targeted Neurons. *Front Comput Neurosci* **14**, 5 (2020).

<https://doi.org/10.3389/fncom.2020.00005>

48. Gysbrechts, B. *et al.* Light distribution and thermal effects in the rat brain under optogenetic stimulation. *J Biophotonics* **9**, 576-585 (2016).

<https://doi.org/10.1002/jbio.201500106>

Reviewer 1: we had already removed the claim, in the previous version of March 7, that “extracting inference requires the organization of previously acquired knowledge into a hierarchical order.” Accordingly, we have modified some sentences in a paragraph titled “Correct performance is observed with incomplete hierarchy setting” (page 7).

Reviewer 2: regarding Figure 3c, we had already replied and modified on March 7 as follows,

☐ In Figure 3c, to avoid confusion and misinterpretation, each session is represented by a distinct color. This color-coding scheme allows readers to easily track the performance of each session throughout the data. We used this color coding across all figures (from Figure 1) for consistency. We have added the following sentence in the revised manuscript (in the legend of Figure 1): “premise pairs were consistently colorcoded across all figures to facilitate tracking each pair without confusion.”

☐ In Figure 3c, the training protocol in (days 11 & 12) is similar to that employed in Figure 1 & 2. However, in days 13 & 14, the 2 pairs were separated to extract specific representations for a particular premise pair without interfering with the other pair.

☐ We have followed the reviewer’s suggestion and have added the following sentence in the revised manuscript (in the legend of Figure 3c): “During training stage, Ca²⁺ transients were collected for each premise pair in a separate day (days 13, 14, 17, 18) to extract specific representations for a particular premise pair without interfering with the other pair.”

2

Reviewer 3: we have revised the discussion section by adding a paragraph regarding both benefits and limitations of using animal model and how does the mouse study presented in this manuscript is related to prior human studies (second paragraph of Discussion starting “Our study demonstrate that offline brain activity”, page 13-14).

A brief summary of the main findings:

It is unknown how sleep creates novel inferential information. Abdou et al. found that neuronal co-activations during NREM sleep organize the learned knowledge in a hierarchy, while co-activations during REM sleep compute the inferential information.

Thank you again for your interest in our work and the time you expended in handling and reviewing our manuscript. We are looking forward to receiving your response.